# Anti-Microbial Activity of Phytocannabinoids and Endocannabinoids in the Light of Their Physiological and Pathophysiological Roles

**DOI:** 10.3390/biomedicines10030631

**Published:** 2022-03-09

**Authors:** Ronit Vogt Sionov, Doron Steinberg

**Affiliations:** The Biofilm Laboratory, The Institute of Biomedical and Oral Sciences, The Faculty of Dentistry, The Hebrew University—Hadassah Medical School, Jerusalem 9112102, Israel; dorons@ekmd.huji.ac.il

**Keywords:** anti-microbial activity, anti-biofilm activity, *Cannabis sativa* L., endocannabinoids, gut microbiota, pathogens, phytocannabinoids

## Abstract

Antibiotic resistance has become an increasing challenge in the treatment of various infectious diseases, especially those associated with biofilm formation on biotic and abiotic materials. There is an urgent need for new treatment protocols that can also target biofilm-embedded bacteria. Many secondary metabolites of plants possess anti-bacterial activities, and especially the phytocannabinoids of the *Cannabis sativa* L. varieties have reached a renaissance and attracted much attention for their anti-microbial and anti-biofilm activities at concentrations below the cytotoxic threshold on normal mammalian cells. Accordingly, many synthetic cannabinoids have been designed with the intention to increase the specificity and selectivity of the compounds. The structurally unrelated endocannabinoids have also been found to have anti-microbial and anti-biofilm activities. Recent data suggest for a mutual communication between the endocannabinoid system and the gut microbiota. The present review focuses on the anti-microbial activities of phytocannabinoids and endocannabinoids integrated with some selected issues of their many physiological and pharmacological activities.

## 1. Introduction

Plant medicine has often been used for the treatment of diverse diseases, including bacterial and fungal infections [1,2,3,4,5,6,7,8]. The plants produce a series of secondary metabolites, many of which have pharmacological as well as anti-microbial activities [4,5,6,9,10,11]. Evolutionarily, plants have developed various anti-microbial mechanisms to protect them from infectious diseases [11]. Usually, these include the production of compounds that have anti-biofilm and bacteriostatic activities rather than biocidal effect [11]. Compounds with anti-biofilm activities are believed not to induce resistance mechanisms in the microbes, since they target processes not essential for their survival. In contrast, compounds with bactericidal activity might lead to the development of resistance mechanisms in the microbe as part of the bacterial fitness adaptation process with increased probability of developing microbial plant infections.

*Cannabis sativa* L. subspecies are plants that contain a large variety of secondary metabolites, including phytocannabinoids, terpenoids and flavonoids, which have profound anti-microbial activities, in addition to possessing anti-inflammatory, anti-oxidative and neuromodulatory properties [12,13,14]. In mammalians, the phytocannabinoids interact with the same receptors (e.g., cannabinoid receptors CB1 and CB2) as the endocannabinoids [15], which are endogenous substances with anti-microbial, anti-inflammatory and neuromodulatory activities [16,17,18,19,20,21,22,23,24]. While much is known about the cannabinoid targets in mammalians, so far, little is known about the microbial targets of these compounds. It is likely that these compounds also interact with specific targets in the microbes. The present review focuses on the anti-microbial activities of phytocannabinoids and endocannabinoids interwoven with selected aspects of their many physiological and pathophysiological activities.

## 2. *Cannabis sativa* L.

The hemp plant (*Cannabis sativa* L.; L = Linnaeus) belonging to the family *Cannabaceae*, originates in central-northeast Asia where it has been cultivated for more than 5000 years [15,25,26]. The Han Chinese dynasty used *Cannabis* to treat inflammatory disorders and malaria [27,28]. The Chinese pharmacopoeia of the Emperor Shen Nung, who lived approximately around 2700 BCE and is considered “The Father of Chinese Medicine”, indicated *Cannabis* plant usage for the treatment of rheumatic pain, constipation, malaria, and gynecological disorders [26]. In modern times, this plant has been used for different medical conditions, including alleviating chronic pain (e.g., in cancer patients and in rheumatic diseases), muscle spasms (e.g., in multiple sclerosis), epileptic convulsion (e.g., Dravet syndrome and Lennox–Gastaut syndrome in children), nausea (e.g., following chemotherapy), intestinal inflammation (e.g., colitis, inflammatory bowel disease (IBD)), and for stimulating appetite (e.g., in devastating AIDS syndrome, anorexia, and cancer patients) [26,29,30]. It has also been used as a treatment remedy for cancer patients, since the phytocannabinoids can inhibit cell growth of certain tumor cells and enhance the efficacy of certain cancer therapeutics [31].

The phenotypes of *Cannabis* plants are highly variable and can be classified into three major subspecies: *Cannabis sativa* subsp. *sativa*, *Cannabis sativa* subsp. *indica*, and *Cannabis sativa* subsp. *ruderalis* [32]. The different subspecies have all been classified to the *Cannabis sativa* L. species [32]. There are also several chemovariants, chemotypes, or cultivars of this plant harboring different composition of chemical compounds [33,34,35,36]. Different *Cannabis* cultivars or chemotypes have been developed that contain various ratios of cannabidiol (CBD) and Δ^9^-tetrahydrocannabinol (Δ^9^-THC), and even those containing high CBD and low Δ^9^-THC content, which is favorable for avoiding the psychomimetic effects of Δ^9^-THC [33,37]. The cannabinoids are found in most parts of the plant, with the highest concentrations in glandular trichomes on the surfaces of leaves and flowers [38,39,40,41,42].

The chemical composition of *Cannabis* is affected by the ripeness and maturation state of the plant, growth conditions, the sowing and the harvest times, as well as the storage conditions [34,38,39,40,41,43]. The plant composition of phytocannabinoids is affected by light, temperature, water supply, nutrition, heavy metals, phytohormones, soil bacteria, insects and microbial pathogens, among others [44,45,46,47]. Cannabidiolic acid (CBDA), the precursor of cannabinols, predominates in the unripen plant, while it is converted to CBD, Δ^9^-THC and cannabinol (CBN) upon ripening of the resin [48]. In the intermediate ripening state, CBD is predominant, then Δ^9^-THC dominates in the ripened state, while CBN, the final conversion product, is the major compound in the overripened resin [48]. High anti-microbial activity was found especially in unripen *Cannabis* harvested from regions with unfavorable climate for this plant, whereas ripened *Cannabis* taken from tropical areas had a more hashish-active composition [48]. For the optimal production of essential oil, the recommended stage for harvest is one to three weeks before seed maturity [43].

The difference between industrial hemp and the high Δ^9^-THC hemp breed type marijuana is that the industrial hemp contains minute amounts of Δ^9^-THC (less than 0.2% (*w*/*v*)), while marijuana flowers and leaves may contain as much as 17–28% Δ^9^-THC [49]. Even concentrated THC products, such as oil, shatter, and dab, have been produced with a concentration of up to 95% Δ^9^-THC [49]. The use of marijuana is associated with hallucinations due to the high Δ^9^-THC content and may lead to addiction, lack of judgement, and reduced cognition, especially during adolescence when the brain is undergoing significant development [49]. Smoking hemp may lead to decreased immune function with a consequent increase in opportunistic infections [50,51,52,53]. *Cannabis* users have a higher probability to get fungal infections than non-*Cannabis* users, which might in part be due to fungal contamination of the *Cannabis* product [54].

### 2.1. Anti-Microbial Activity of Cannabis sativa *L.* Extracts

Z. Krejčí, in the 1950s, observed that *Cannabis* has antibiotic activity and introduced it to the clinics in Czechoslovakia [55], a practice that was discontinued in 1990 [33]. The first compound identified by Krejčí with antibiotic activity was named cannabidiolic acid (CBDA) [56,57]. From then on, several other *Cannabis* components with antibiotic activities have been isolated and characterized [48,58,59,60,61,62,63], which will be further discussed below. In 1956, L. Ferenczy published a paper documenting that plant seeds from various plant species, including those from *Cannabis sativa*, exhibited antibacterial activity, especially against Gram-positive bacteria [64]. Wasim et al. [65] documented that both ethanolic and petroleum ether extracts of *Cannabis sativa* leaves showed anti-microbial activity against *Bacillus subtilis*, *Staphylococcus aureus*, *Micrococcus flavus*, *Bordetella bronchiseptica*, *Proteus vulgaris*, *Aspergillus niger*, and *Candida albicans*. Ali et al. [66] observed that the oil of the seeds of *Cannabis sativa* exerted pronounced anti-bacterial activity against *Bacillus subtilis* and *Staphylococcus aureus*, with moderate activity against *Escherichia coli* and *Pseudomonas aeruginosa*, without any activity against *Aspergillus niger* and *Candida albicans*. The petroleum ether extract of the whole plant showed high anti-bacterial activity against *Bacillus subtilis* and *Staphylococcus aureus*, moderate activity against *Escherichia coli*, while no activity against *Pseudomonas aeruginosa* or the tested fungi [66]. Thus, the extraction method and the source affect the composition of the anti-microbial content and the spectrum of responding microbes.

### 2.2. Anti-Microbial Activity of Essential Oils from Cannabis sativa *L.*

Novak et al. [67] analyzed the anti-bacterial effect of essential oils prepared from five different cultivars of *Cannabis sativa* L. These essential oils contained, among others, α-pinene, myrcene, trans-β-ocimene, α-terpinolene, trans-caryophyllene, and α-humulene, but undetectable levels of Δ^9^-THC and very poor levels of other cannabinoids [67]. They observed differences in the anti-bacterial activity between the various cultivars. All five essential oils showed anti-bacterial activity against *Acinetobacter calcoaceticus*, *Beneckea natriegens*, *Brochothrix thermosphacta* and *Staphylococcus aureus* [67]. Only one of the five essential oils had an anti-bacterial effect on *Escherichia coli*, while none affected *Enterobacter aerogenes*, *Klebsiella pneumoniae*, *Proteus vulgaris*, *Salmonella pullorum*, *Serratia marcescens*, or *Streptococcus faecalis* [67].

Nissen et al. [34] observed that essential oils of *Cannabis sativa* L., prepared from 50–70% of seed maturity, showed anti-bacterial activity against the Gram-positive bacteria *Enterococcus faecium* and *Streptococcus salivarius* at less than 1% (*v*/*v*) but were unable to inhibit the growth of the yeast *Saccharomyces cerevisiae*. Zengin et al. [68] found that essential oils distilled from leaves, inflorescences, and thinner stems of the hemp plant showed anti-oxidative properties and had significant anti-bacterial activity against clinical *Helicobacter pylori* strains (MIC = 16–64 μg/mL), with lower activity against clinical *Staphylococcus aureus* isolates (MIC = 8 mg/mL) and no significant activity against *Candida* spp. and *Malassezia* spp. The minimum bacterial biofilm inhibitory concentration (MBIC) of the hemp essential oil against *Helicobacter pyroli* was similar to the MIC [68]. The hemp essential oil showed cytotoxicity against human breast cancer, cholangiocarcinoma, and colon carcinoma cell lines at 50–75 μg/mL, while 250 μg/mL was required to inhibit the cell proliferation of a nonmalignant cholangiocyte cell line [68]. The LD_50_ of hemp essential oil against larvae of *Galleria mellonella* was found to be 1.56 mg/mL, which is much higher than the anti-bacterial activity against *Helicobacter pyroli*, but lower than that found to be active against *Staphylococcus aureus* strains [68].

Pellegrini et al. [69] observed that essential oil prepared from *Cannabis sativa* L. cultivar Futura 75 inflorescences with low Δ^9^-THC content (<0.2%) cultivated in the Abruzzo territory showed anti-bacterial activity against *Staphylococcus aureus* and *Listeria monocytogenes* with a MIC of 1.25–5 µL/mL, while being ineffective against *Salmonella enterica*. They also showed that the essential oil possessed anti-oxidative properties [69]. The essential oils produced from the *Cannabis sativa* L. cultivar Futura 75 inflorescences was also found to have insecticidal activity with LD_50_ values of 65.8 μg/larva on *Spodoptera littoralis*, 122.1 μg/adult on *Musca domestica*, and LC_50_ of 124.5 μL/L on *Culex quinquefasciatus* larvae [70]. The insecticidal effect might in part be due to an inhibition of the enzyme acetylcholinesterase (AChE) [70]. Thomas et al. [71] found that essential oil of *Cannabis sativa* could induce 100% mortality in the mosquito larvae of *Culex tritaeniorhynchus*, *Anopheles stephensi*, *Aedes aegypti*, and *Culex quinquefasciatus* at concentrations of 0.06, 0.1, 0.12, and 0.2 μL/mL, respectively.

Palmieri et al. [72] studied the variability of *Cannabis* essential oils from various origins and observed that the time of distillation affected the chemical composition of terpenic components, sesquiterpenes, and CBD with consequent variations in the anti-microbial activities against *Staphylococcus aureus*, *Listeria monocytogenes*, and *Enterococcus faecium*. Zheljazkov et al. [73] compared the anti-microbial activity of nine wild hemp (*Cannabis sativa* spp. *spontanea* Vavilov) accessions sampled from agricultural fields in northeastern Serbia with 13 EU registered cultivars, eight breeding lines, and one cannabidiol (CBD) hemp strain, which showed variations in the secondary metabolites β-caryophyllene, α-humulene, caryophyllene oxide, and humulene epoxide. The CBD concentration in the essential oils of wild hemp varied from 6.9 to 52.4%, while the CBD content in the essential oils of the registered cultivars, breeding lines, and the CBD strain varied from 7.1 to 25%; 6.4 to 25%; and 7.4 to 8.8%, respectively [73]. The Δ^9^-THC concentration showed high variability between the different strains, with the highest concentration being 3.5% [73]. The essential oils of the wild hemp had greater anti-microbial activity compared with the essential oil of registered cultivars [73]. In general, with variations between the different essential oils, anti-microbial activity was observed toward *Staphylococcus aureus*, *Enterococcus faecalis*, *Streptococcus pneumoniae*, *Pseudomonas aeruginosa*, *Yersenia enterocolitica*, *Salmonella enterica*, *Candida albicans*, *Candida krusei*, and *Candida tropicalis* using the disc diffusion method [73]. Altogether, the data presented above show that there is high variability of the composition of hemp essential oils, which might explain the many contradictory publications of the anti-microbial activities toward the same microbial species. In general, a good anti-bacterial response is achieved on Gram-positive bacteria, with less or no effect on Gram-negative bacteria, and variable effect on fungi.

### 2.3. Anti-Microbial Activity of Terpenoids in Cannabis Essential Oils

Several terpenoids in the *Cannabis* essential oils have been demonstrated to have anti-microbial effect, which include the monoterpenes α-pinene, linalool, and limonene, and the bitter-tasting sesquiterpenes nerolidol, β-caryophyllene, and caryophyllene oxide [33,74,75,76]. α-Pinene inhibited the growth of both Gram-positive bacteria (e.g., various *Clostridium* species, *Enterococcus faecium*, *Streptococcus salivarius*, *Staphylococcus aureus*, *Staphylococcus epidermidis*, *Streptococcus pyogenes*, *Streptococcus pneumoniae*) and Gram-negative bacteria (e.g., various *Pseudomonas* species), as well as the fungus *Candida albicans* [34,77,78,79]. Myrcene, which is also found in tea tree oil, inhibited the growth of *Staphylococcus aureus* that was associated with the leakage of K^+^ ions from the bacterial cells and damage to the cell membrane [80]. Linalool, a monoterpenoid alcohol, and α-terpineol, a fragrant terpene, showed anti-bacterial activity against *Propionibacterium acne* and *Staphylococcus epidermidis* with a minimum inhibitory concentration (MIC) of 0.625–1.25 µg/mL [77]. Linalool is also effective against the yeast and hyphal forms of *Candida albicans*, where it alters the membrane integrity and induces cell cycle arrest [81]. Limonene showed anti-bacterial activity against *Staphylococcus epidermidis* [77] and *Listeria monocytogenes* [82], and exerted anti-biofilm activity against *Streptococcus pyogenes*, *Streptococcus mutans*, and *Streptococcus mitis* [83]. α-Humulene showed potent anti-fungal activity against *Cryptococcus neoformans*, *Candida glabrata*, and *Candida krusei* with MIC values of 5.0, 1.45, and 10.0 μg/mL, respectively, without any effect on methicillin-sensitive *Staphylococcus aureus* (MSSA) 29213, methicillin-resistant *Staphylococcus aureus* (MRSA) 33591, or *Mycobacterium intracellulare* [84]. Nerolidol is a sesquiterpene with sedative properties and inhibits the growth of *Leishmania amazonensis*, *Leishmania braziliensis*, and *Leishmania chagasi* promastigotes, and *Leishmania amazonensis* amastigotes [85], as well as the growth of *Plasmodium falciparum* at the trophozoite and schizont stages [86,87]. The anti-oxidative β-caryophyllene possesses anti-microbial activity against *Staphylococcus aureus* (MIC 2–4 µM), *Bacillus subtilis* (MIC 6–10 µM), *Escherichia coli* (MIC 7–11 µM), *Pseudomonas aeruginosa* (6–8 µM), *Aspergillus niger* (MIC 5–7 µM), and *Trichoderma reesei* (MIC 3–5 µM) without any significant cytotoxic effect on normal mammalian cell lines [88]. The anti-inflammatory oxygenated sesquiterpene caryophyllene oxide exhibited anti-fungal activities against the dermatophytes *Trichophyton mentagrophytes* var. *mentagrophytes*, *Trichophyton mentagrophytes* var. *interdigitale*, and *Trichophyton rubrum* [89].

## 3. Phytocannabinoids

The *Cannabis sativa* L. plants produce more than 560 chemicals, including at least 144 cannabinoids and 200 terpenoids, as well as flavonoids and polyunsaturated fatty acids [15,33,34,42,63,67,72,73,90,91,92,93,94,95,96,97,98,99,100,101,102,103,104,105,106,107]. The most common phytocannabinoids are Δ^9^-tetrahydrocannabinol (Δ^9^-THC) and cannabidiol (CBD), which are the neutral homologs of tetrahydrocannabinolic acid (THCA) and cannabidiol acid (CBDA), respectively [108]. The phytocannabinoids are terpenophenolic compounds containing a resorcinyl core with a para-positioned isoprenyl, alkyl, or aralkyl side chain [39,40] (Figure 1). The tetrahydrobenzochromen ring is quite unique to the genus *Cannabis*, although a related compound has been found in the liverwort *Radula marginata* [109], and cannabigerol (CBG) and its corresponding acid have been isolated from *Helichrysum umbraculigerum* [110].

Apart from exerting anti-microbial activities, which will be discussed in more detail below (Section 3.3), phytocannabinoids modulate several physiological and pathophysiological processes in humans and other mammalians, making them potential therapeutic drugs in various settings [12,13,14,31,111,112,113,114,115]. Among others, these compounds have been shown to have anti-inflammatory, anti-oxidative, anti-nausea, anti-nociceptive, anti-convulsant, anti-neoplastic, anxiolytic, and neuroprotective properties [14,111,112,114,115,116,117]. Cannabinoids also affect cognition, such as learning and memory, consciousness, and emotion, including anxiety and depression [118,119].

Some cannabinoid-based drugs (e.g., Marinol, Syndros, Cesamet, Sativex, and Epidiolex) have been approved by the U.S. Food and Drug Administration (FDA) for the treatment of epilepsy, Dravet syndrome, Lennox–Gastaut syndrome, Parkinson’s disease, spasticity associated with multiple sclerosis, neuropathic pain, mental illnesses, chemotherapy-induced nausea, and AIDS wasting syndrome [117,120,121,122]. Marinol and Syndros contain the (-)-trans-Δ^9^-THC dronabinol; Cesamet contains the synthetic cannabinoid nabilone that shows structural similarities to Δ^9^-THC; and Epidiolex contains CBD. Sativex is produced from a *Cannabis*-derived extract that is composed of approximately equal quantities of Δ^9^-THC and CBD. A major concern is the production of many psychotropic synthetic cannabinoids distributed on the illicit market, which poses a potential health treat due to their high potency and toxicity [123].

### 3.1. Cannabinoid Receptors

The effects of phytocannabinoids on humans and other mammalians are partly mediated by the G_i/o_ protein-coupled CB1 (encoded by the CNR1 gene) and CB2 (encoded by the CNR2 gene) cannabinoid receptors that consist of seven transmembrane domains [124,125,126]. The stimulation of these receptors leads to the inhibition of adenylyl cyclase with consequent reduction in the intracellular cAMP levels, activation of potassium channels, activation of mitogen-activated protein kinases (MAPKs) such as the extracellular signal-regulated kinase (ERK) and c-Jun *N*-terminal kinase (JNK), as well as activation of the phosphoinositide-3 kinase (PI3K)/Akt signaling pathways and the mammalian target of rapamycin (mTOR) [126,127,128,129,130,131,132,133,134].

The CB1 and CB2 receptors also recognize the endogenous arachidonic acid-derived endocannabinoids, such as *N*-arachidonoylethanolamine (anandamide; AEA) and 2-arachidonoylglycerol (2-AG) [134,135,136]. Both CB1 and CB2 are expressed in various cells in the brain and in peripheral tissues [137]. CB1 is especially expressed at high levels in the neocortex, hippocampus, basal ganglia, cerebellum, and brainstem, but it is also found in peripheral nerve terminals and some tissues, such as the vascular endothelium, spleen, testis, and eye [137]. CB2 is predominantly found in cells of the immune system, and in the central nervous system, it is primarily localized to microglia and tissue macrophages [137].

The CB1 receptor regulates the balance between excitatory and inhibitory neuronal activity. The psychoactive effect is believed to be mediated through the CB1 receptor in the brain, whereas the immunomodulatory effects are anticipated to be mediated via the CB2 receptor expressed on immune cells [138,139]. In addition, CB1 signaling affects metabolism and is involved in maintaining whole body energy homeostasis by increasing appetite and stimulating feeding [140]. Many efforts have been made to develop CB2 specific agonists at an attempt to achieve anti-inflammatory actions without psychotropic adverse effects [13,141,142,143]. The sesquiterpene (E)-β-caryophyllene produced by *Cannabis* as well as other plants, including oregano (*Origanum vulgare* L.), cinnamon (*Cinnamomum* spp.), and black pepper (*Piper nigrum* L.), was found to bind selectively to the CB2 receptor and exert anti-inflammatory activities [144,145,146,147].

Other cannabinoid receptors include transient receptor potential vanilloid 1 (TRPV1), the G-protein-coupled receptors GPR18 and GPR55, and peroxisome proliferator activated receptors (PPARs) [126,134,148,149,150,151,152]. The anti-nociceptive effect of *Cannabis sativa* extracts was found to be mediated by the binding of CBD to TRPV1 [153]. A study by Ibrahim et al. [154] showed that activation of the CB2 receptor by its agonist AM1241 stimulated the release of beta-endorphin from keratinocytes, which, in turn, acted on neuronal μ-opoid receptors to inhibit nociception. The *Cannabis sativa* extract containing multiple cannabinoids, terpenes, and flavonoids had stronger anti-nociceptive effect than a single cannabinoid given alone [153], suggesting an “entourage” effect of the various *Cannabis*-containing compounds [74].

The CB1 and CB2 can form receptor heteromers [155]. The activity of the receptor heteromer is affected by the agonists and antagonists that bind to each of them. A CB1 antagonist can block the effect of a CB2 agonist and vice versa; a CB2 antagonist can block the effect of a CB1 receptor agonist [155]. CB1 has also been shown to form heteromers with dopamine and adenosine receptors [156,157,158], AT1 angiotensin receptor [159], μ_1_-opoid receptor [160,161], and OX1 orexin A receptor [162]. The many interacting partners put CB1 signaling under strict regulation.

### 3.2. Pharmacological Effects of Selected Phytocannabinoids

#### 3.2.1. Δ^9^-Tetrahydrocannibinol (Δ^9^-THC)

Δ^9^-THC binds to CB1 and CB2 receptors at a more or less equal affinity [138,163,164]. It also acts on CB1-CB2 receptor heterodimers [165]. Δ^9^-THC is well known for its psychomimetic activities that are exerted by its binding to CB1 receptor in the brain, resulting in a calm and sedated mental state [49]. Besides euphoria, Δ^9^-THC is an appetite stimulator [166]. Oral Δ^9^-THC (Dronabinol, Marinol) and its synthetic nabilone (Cesamet) have been used for the treatment of nausea and appetite stimulation for people undergoing chemotherapy and for AIDS wasting syndrome [167,168]. The activation of CB1 by Δ^9^-THC is believed to mediate its anti-nausea and anti-emetic effects [169]. Sativex, which contains a combination of Δ^9^-THC and CBD, has been used for relief of neuropathic pain in multiple sclerosis [170].

#### 3.2.2. Cannabidiol (CBD)

The non-psychotropic cannabidiol (CBD) shows low affinity to the CB1 and CB2 receptors [135] and can exert antagonistic modulatory actions on these receptors [138,171]. CBD can also activate the TRPV1 channel, serotonin 1A (5-HT_1A_) receptors, and opioid receptors [24,172]. CBD has anti-inflammatory, anti-oxidative, anti-epileptic, analgesic, anti-neoplastic, sedative, neuroprotective, and anti-anxiety activities [173,174,175,176,177,178,179,180,181,182,183,184,185,186,187,188]. Moreover, CBD inhibits sebocyte lipogenesis by activating the TRPV4 ion channel that interferes with the pro-lipogenic ERK1/2 MAPK pathway [189].

The neuroprotective activity of CBD has been attributed in part to its anti-oxidative activity [190,191]. Based on its immunomodulatory activities, CBD has been implicated in the treatment of various autoimmune diseases [14,21], and its anti-nociceptive activity was found to be beneficial in relieving chronic pain [192]. In addition, CBD has potential uses in psychiatry due to its neuromodulatory activities in the brain that control recognition, emotional and behavioral responses [111,193,194]. CBD has especially been reported to have therapeutic effect for psychopathological conditions, such as substance use disorders, chronic psychosis, and anxiety [193]. CBD has been shown to be well tolerated in humans at concentrations as high as 3500–6000 mg/day [195,196,197], and the FDA-approved CBD (marketed as Epidiolex) is indicated for preventing epileptic seizures in Lennox–Gastaut syndrome and Dravet syndrome in children [198].

In experimental mice and rat models, CBD has been shown to have immunosuppressive activities [181], which are partly due to inhibition of TNFα production [199,200] and induction of myeloid-derived suppressor cells (MDSCs) [201]. CBD alleviated the symptoms of experimental autoimmune encephalomyelitis (EAE) and collagen-induced arthritis and prevented the onset of autoimmune diabetes in experimental murine models [199,200,202]. In mice, the anti-inflammatory activity of CBD was found to have a bell-shaped dose–response with an optimal dose of 5 mg/kg [203]. The use of a standardized extract from a CBD-rich, ∆^9^-THC^low^ *Cannabis indica* cultivar overcame this bell-shaped dose–response, suggesting a synergistic effect among the different compounds of the *Cannabis* extract [199].

#### 3.2.3. Cannabigerol (CBG)

CBG is another non-psychoactive *Cannabis* component that is produced at elevated levels in some industrial hemps [204,205,206]. It binds to both CB1 and CB2 receptors and modulates the signaling through these receptors, as well as the CB1-CB2 receptor heteromer, at concentrations as low as 0.1–1 μM [207]. CBG competes with the binding of [^3^H]-WIN-55,212-2 to CB2, but not to CB1 [207]. Further studies suggest that CBG is a partial agonist of CB1 and CB2 [207,208,209]. CBG activates TRPV1, TRPV2, TRPV3, TRPV4, TRPA1, 5-HT_1a_ receptor, α2-adrenergic receptor, and PPARγ, while being a TRPM8 antagonist [210,211,212,213,214,215]. CBG has anti-inflammatory, anti-oxidative, and anti-nociceptive activities [117,209,213,216]. The anti-inflammatory property is thought to be achieved by modulating the CB2 receptor, TRP channels, and PPARγ, and by inhibiting cyclooxygenase 1 and 2 (COX-1/2) [210,211,217], while the analgesic effect of CBG is thought to be mediated through the α2-adrenergic receptor [211]. CBG has been shown to have potential beneficial effects in treating inflammatory bowel disease and neurological disorders, such as Huntington’s disease, Parkinson’s disease, and multiple sclerosis [213,215,216,218,219].

#### 3.2.4. Cannabichromene (CBC)

CBC is a non-psychoactive phytocannabinoid that activates the CB1 and CB2 receptors, resulting in decreased intracellular levels of cAMP [209]. CBC also activates the TRPA1, TRPV3, and TRPV4 channels [210]. CBC has anti-inflammatory, anti-nociceptive, and neuroprotective activities [220,221,222,223,224,225]. CBC reduces the activity of both the ON and OFF neurons in the rostral ventromedial medulla (RVM) and elevates the endocannabinoid levels in the ventrolateral periaqueductal gray matter [221]. The anti-nociceptive activity of CBC is mediated by the adenosine A1 and TRPA1 receptors [221]. CBC increases the viability of neural stem progenitor cells through activation of the adenosine A1 receptor [224]. Moreover, it has been shown to suppress reactive astrocytes, thus offering a protective effect against neuro-inflammation and Alzheimer’s disease [225]. CBC had anti-convulsant properties in a mouse model of Dravet syndrome [226], and it exhibited cytotoxic activity against some carcinoma cells [227,228].

#### 3.2.5. Cannabidiolic Acid (CBDA)

CBDA has low affinity for both CB1 and CB2 receptors, with moderate inhibition of adenylyl cyclase activity [209,229], and functions as an allosteric regulator on the 5-HT_1A_ receptor, resulting in anti-emetic effects [230,231,232,233]. In addition, it activates PPARα and PPARγ [212]. CBDA shows anti-nociceptive and anti-inflammatory effects that are in part mediated by COX-2 inhibition and activation of the TRPV1 channel [217,234,235]. CBDA has anxiolytic and anti-convulsant effects in animal models [236,237,238].

#### 3.2.6. Cannabigerolic Acid (CBGA)

CBGA displays low affinity for both CB1 and CB2 receptors but causes a similar decrease in intracellular cAMP levels as Δ^9^-THC [229]. Since CBGA can activate PPARs [212], it is expected to affect lipid metabolism [117]. A *Cannabis sativa* cultivar containing high levels of CBG and CBGA inhibited the activity of the aldose reductase enzyme, which catalyzes the reduction of glucose to sorbitol [239]. Since the aldose reductase level is increased at high glucose levels and has been implicated in the development of neuropathy, nephropathy, retinopathy, and cataract in diabetes, CBGA has been suggested as a potential drug in preventing diabetic complications [239]. In the Scn1a^+/−^ mouse model of Dravet syndrome, CBGA was found to have an anti-convulsant effect that was mediated by its interaction with the GPR55, TRPV1, and GABA_A_ receptors [240].

#### 3.2.7. Cannabinol (CBN)

CBN is formed during the degradation of Δ^9^-THC and has a lower binding affinity to CB1 and CB2 receptors than Δ^9^-THC [117]. CBN is an agonist of the TRPV1, TRPV2, TRPV3, TRPV4, and TRPA1 cation channels [210]. CBN is a non-psychotropic phytocannabinoid with analgesic and anti-inflammatory properties and acts as an appetite stimulant [117]. CBN has neuroprotective activity that is associated with its anti-oxidative actions, trophic support, and elimination of intraneuronal β-amyloid in neuronal cells [241]. CBN preserves mitochondrial functions, such as redox regulation, calcium uptake, mitochondrial membrane potential, and bioenergetics [242]. CBN promotes endogenous antioxidant defense mechanisms and triggers AMP-activated protein kinase (AMPK) signaling pathways [242].

### 3.3. Anti-Microbial Effects of Phytocannabinoids

Several phytocannabinoids have been shown to have anti-bacterial activities, especially on Gram-positive bacteria, including various antibiotic-resistant strains [58,59,62,63,101,220,243,244,245,246,247] (Table 1). Phytocannabinoids have been shown to exert both bactericidal and bacteriostatic effects [61,62,244,247]. Most of the studies have analyzed the half maximal inhibitory concentration (IC_50_) or minimum inhibitory concentration (MIC) for each of the compounds against different bacterial species, fungi, and protozoa, while only a few studies have looked at the underlying mechanisms [61,243,244,247,248,249,250] (Figure 2).

#### 3.3.1. Bacterial Growth Inhibitory Effects of Phytocannabinoids

The minimum inhibitory concentration (MIC) of Δ^9^-THC and CBD on various *Staphylococcus aureus* strains, including MRSA and *Streptococci* species (e.g., *Streptococcus pyogenes* and *Streptococcus. faecalis*) was found to be in the range of 1–5 μg/mL [58,62,245,246]. There was no significant difference between the anti-bacterial effect of Δ^9^-THC and CBD [58,245,246]. The anti-microbial effect was attenuated by the presence of either serum or blood, suggesting that serum components can bind the compounds and prevent them from acting on the microorganisms [245]. CBG shows anti-bacterial activity against Gram-positive bacteria, including MSSA, MRSA, and the oral cariogenic *Streptococcus mutans* at low concentrations similar to CBD [58,61,244,247]. CBC and CBDA showed a MIC of 1–2 μg/mL against *Staphylococcus aureus* and *Staphylococcus epidermidis* [62,220]. In these studies, CBDA was less active than CBD [62]. Cannabichromenic acid (CBCA) caused a rapid reduction in the colony-forming units (CFUs) of a clinical MRSA isolate both during the exponential and stationary growth phase, suggesting a bactericidal activity that is independent of the metabolic state of the bacteria [254]. None of the phytocannabinoids had any significant anti-bacterial activity against Gram-negative bacteria, such as *Escherichia coli*, *Salmonella typhi*, *Pseudomonas aeruginosa*, and *Proteus vulgaris* [61,62,220,245,247]. This might be due to the inability of these compounds to penetrate the outer membrane of the Gram-negative bacteria [61], or the outer membrane protects the bacteria from cell death caused by damage to the inner membrane.

#### 3.3.2. Outer Membrane Permeabilization of Gram-Negative Bacteria Sensitizes Them to Phytocannabinoids

Interestingly, CBD and CBG could act on some Gram-negative bacteria (e.g., *Escherichia coli*, *Acinetobacter baumannii*, *Klebsiella pneumoniae*, *Pseudomonas aeruginosa*) if the outer membrane was permeabilized with the LPS-binding antibiotic polymyxin B [61,247]. It was shown that an *Escherichia coli* Δ*bamB*Δ*tolC* deletion strain that renders the bacteria hyperpermeable to many small molecules was sensitive to CBG with a MIC of 4 μg/mL, which is in contrast to the parental *Escherichia coli* wild-type strain that showed a MIC above 128 μg/mL [61]. Similarly, a lipo-oligosaccharide-deficient *Acinetobacter baumannii* strain became sensitive to CBG with a MIC of 0.5 μg/mL compared to the parental strain showing a MIC of 64 μg/mL [61].

#### 3.3.3. Combined Treatment of Phytocannabinoids with Antibiotics

No synergistic or antagonistic effects of CBD were observed on MRSA strain USA300 when combined with different conventional antibiotics, such as clindamycin, ofloxacin, meropenem, tobramycin, methicillin, teicoplanin, and vancomycin [62]. These authors concluded that the membrane-perturbing effect of CBD was not sufficient to enhance the uptake of conventional antibiotics [62]. However, Wassmann et al. [251] observed that CBD could reduce the MIC value of bacitracin against several Gram-positive bacteria, including *Staphylococcus* species, *Listeria monocytogenes*, and *Enterococcus faecalis*. The simultaneous use of CBD and bacitracin on MRSA USA300 resulted in the formation of multiple septa during cell division, appearance of membrane irregularities, reduced autolysis, and decreased membrane potential [251]. The combined CBD/bacitracin treatment did not affect the growth of the Gram-negative bacteria *Pseudomonas aeruginosa*, *Salmonella typhimurium*, *Klebsiella pneumoniae*, and *Escherichia coli* [251].

#### 3.3.4. Phytocannabinoids Also Act on Persister Cells and Do Not Induce Drug Resistance

CBG was found to be active against MRSA persister cells, which are dormant, non-dividing bacteria [61]. This trait is therapeutically important, since many antibiotics require cell division to be effective, and they are frequently unable to eradicate persister cells that usually recover after antibiotic withdrawal [255,256,257]. Another obstacle of antibiotic therapy is the development of drug resistance, a frequent reason for treatment failure [258]. Farha et al. [61] attempted to develop CBG-resistant bacteria in the hopes of finding the target molecules. Despite rechallenging the MRSA with 2x and 16x MIC concentration of CBG, they were unable to get any spontaneously CBG-resistant mutants [61]. Similarly, MRSA that had been daily exposed to sub-lethal concentration of CBD for 20 days were still sensitive to CBD [247]. The authors of these two studies [61,247] concluded that CBD and CBG do not induce drug resistance. However, it should be noted that following exposure to CBD or CBG, the surviving growth-arrested bacteria could regain growth after withdrawal of the drug.

#### 3.3.5. Therapeutic Anti-Microbial Potential of Phytocannabinoids

The hemolytic activity of CBD and CBG was found to be 256 μg/mL and 32 μg/mL, respectively, which is far above the MIC of 1–4 μg/mL for MRSA [61,247]. Additionally, the hemolytic activity of CBDA was found to be above 32 μg/mL [62]. This makes phytocannabinoids potential drugs that can act within a reasonable therapeutic window.

Farha et al. [61] observed that treating MRSA-infected mice with a high dose of 100 mg/kg CBG could reduce the bacterial burden in the spleen by a 2.8 log_10_ of CFU. Blaskovich et al. [247] tried various CBD-containing ointment formulations that could reduce a 2–3 log_10_ of CFU of MRSA inoculated on porcine skin after 1 h and a reduction of more than 5 log_10_ of CFU after a 24 h incubation. CBD, however, failed to significantly reduce the bacterial load of MRSA ATCC 43300 in a thigh infection mouse model [247].

#### 3.3.6. Anti-Biofilm Activities of Phytocannabinoids

Biofilms are communities of bacteria embedded in an extracellular matrix that have attached to a biotic surface (e.g., lung tissue, gastrointestinal tract, nasal mucosa, inner ear) or an abiotic surface (e.g., medical devices, such as catheters, heart valves, stents, prostheses) [259]. The majority of infectious diseases involve bacterial biofilms that are usually difficult to eradicate due to reduced antibiotic sensitivity [259,260]. Several studies show that CBD and CBG can prevent biofilm formation of various Gram-positive bacteria (e.g., MSSA, MRSA, *Streptococcus mutans*) [61,243,247]. The extent of anti-biofilm activity of CBD and CBG against these bacteria correlated with their anti-bacterial activity [61,243,244,247]. In most cases, a similar concentration of these compounds was required to achieve both effects, suggesting that some of the anti-biofilm effect is caused by the anti-bacterial activity [61,243,244]. Moreover, CBD was found to be able to eradicate preformed MSSA and MRSA biofilms with a minimum biofilm eradication concentration (MBEC) of 1–4 μg/mL, indicating that CBD can penetrate the biofilms and act on the biofilm-embedded bacteria [247]. Some cannabinoids (e.g., CBD, CBG, CBC, and CBN) were shown to reduce the bacterial content of dental plaques in an in vitro assay where dental plaques were spread on agar plates coated with the cannabinoids [261]. The anti-biofilm activity of the cannabinoids has significant clinical importance, since the bacteria-embedded bacteria frequently show antibiotic resistance, and some antibiotics are unable to penetrate through the extracellular matrix of the biofilms [259,262,263].

#### 3.3.7. Anti-Fungal Biofilm Activities of Phytocannabinoids

CBD barely affects the viability of *Candida albicans* with a MIC above 50–100 µg/mL [247,253], but it reduces biofilm formation with a biofilm inhibitory concentration 50 (BIC_50_) at 12.5 µg/mL and a MBIC_90_ of 100 µg/mL [253]. CBD reduced the metabolic activity of preformed *Candida albicans* biofilms by 50–60% at 6.25 µg/mL with no further reduction at higher concentrations, even at 100 µg/mL [253]. The morphology of the *Candida albicans* biofilm becomes altered in the presence of CBD. While the hyphal form was predominant in control biofilms, the CBD (25 µg/mL)-treated biofilms appeared in clusters mostly in yeast and pseudohyphal forms [253]. CBD caused a dose-dependent reduction in the cell wall chitin content and the intracellular ATP level, while increasing the intracellular reactive oxygen species (ROS) levels [253]. Gene expression studies showed that after a 24 h incubation with 25 µg/mL CBD, there is a significant downregulation of: *ADH5* (Alcohol dehydrogenase 5), involved in extracellular matrix production; *BIG1*, required for synthesis of the extracellular matrix component β-1,6-glucan; *ECE1* (extent of cell elongation protein 1), involved in biofilm formation; *EED1*, involved in filamentous growth; *CHT1* and *CHT3* chitinases, involved in the remodeling of chitin in the fungal cell wall; and *TRR1* (thioredoxin reductase) with anti-oxidant properties. On the other hand, a significant upregulation of *YWP1* (yeast-form wall protein 1) which is expressed predominantly in the yeast form, was observed [253]. These changes in gene expression might explain, at least in part, the reduced biofilm mass of *Candida albicans* in the presence of CBD and the increase in oxidative stress [253].

#### 3.3.8. Anti-Viral Activities of Phytocannabinoids

There are some lines of evidence for an anti-viral activity of phytocannabinoids [60,264]. Some phytocannabinoids, especially Δ^9^-THC and CBD, bind to the M^pro^ protease of SARS-CoV-2, which plays a role in viral replication [60,264]. CBGA and CBDA were found to be allosteric and orthosteric ligands for the spike protein of SARS-CoV-2 and prevented infection of human epithelial cells by a pseudovirus expressing the SARS-CoV-2 spike protein [265]. Phytocannabinoids might indirectly relieve the disease progress of COVID-19 patients through their anti-inflammatory properties [266]. However, CBD failed to alter the clinical disease development of COVID-19 when given at a daily dose of 300 mg for 14 days [267]. Additionally, caution should be taken into account due to the immunosuppressive activities of phytocannabinoids that can prevent proper anti-viral immune responses [268]. Notably, the use of *Cannabis* was increased in U.S. and Canada by 6–8% during the COVID-19 pandemic in comparison to the pre-pandemic period [269], with a special increase among people with mental health [270]. Vulnerability to COVID-19 was correlated with genetic liability to *Cannabis* use disorder (CUD) [271].

### 3.4. Some Mechanistic Insight into the Anti-Bacterial Activity of Phytocannabinoids

The ability of phytocannabinoids such as CBD and CBG, to kill MRSA, NorA-overexpressing *Staphylococcus aureus*, vancomycin-resistant *Staphylococcus aureus* (VRSA), vancomycin-resistant *enterococci* (VRE) to a similar extent as the respective antibiotic-sensitive strains [58,245,247], suggests that its action mechanism is not hindered by the common antibiotic-resistance mechanisms. Thus, phytocannabinoids can be used as an alternative drug or an antibiotic adjuvant for infectious diseases caused by drug-resistant Gram-positive bacteria.

#### 3.4.1. CBD and CBG Target the Cytoplasmic Membrane, Increase Membrane Permeability, and Reduce Metabolic Activity

There is evidence that CBD and CBG act by targeting the cytoplasmic membrane of the Gram-positive bacteria [61,247]. Exposure of MSSA and MRSA to CBD or CBG caused a dose-dependent increase in the fluorescence of the potentiometric probe 3,3′-dipropylthiadicarbocyanine iodide [DiSC3(5)], suggesting a CBG-induced membrane depolarization [61,247]. CBD inhibited protein, DNA, RNA, and peptidoglycan synthesis in a *Staphylococcus aureus strain* when using concentrations close to the MIC [247]. At sub-MIC levels, CBD inhibited lipid synthesis [247]. CBG was found to inhibit the enzyme enoyl acyl carrier protein reductase (InhA) [272], which is involved in type II fatty acid biosynthesis in *Mycobacterium tuberculosis*. The rapid uptake of the SYTOX green dye into *Staphylococcus aureus* and *Bacillus subtilis* by CBD at MIC, suggests that CBD causes an increase in membrane permeability [247].

CBG prevents the growth of oral cariogenic *Streptococcus mutans* in a concentration and bacterial cell density manner [243]. At a MIC of 2.5 μg/mL, CBG exhibited a bacteriostatic effect on *Streptococcus mutans*, while at 2x MIC and 4x MIC, a bactericidal activity was observed [243]. CBG treatment was found to alter the morphology of *Streptococcus mutans* and cause intracellular accumulation of membrane-like structures [243]. CBG induced an immediate membrane hyperpolarization, followed by increased uptake of propidium iodide, suggesting increased membrane permeabilization [243]. At the same time, Laurdan incorporation into the membranes was reduced in a dose-dependent manner [243], indicative of a more rigid membrane structure. The metabolic activity was decreased in a dose-dependent manner, which might contribute to the growth inhibitory effect [243].

#### 3.4.2. CBD Inhibits the Release of Membrane Vesicles from *Escherichia coli*

Kosgodage et al. [250] observed that CBD inhibits the release of membrane vesicles from the Gram-negative *Escherichia coli* VCS257, while having negligible effect on the membrane vesicle release from the Gram-positive *Staphylococcus aureus* subsp. *aureus* Rosenbach. Membrane vesicles participate in inter-bacterial communication by the transfer of cargo molecules and virulence factors [273]. CBD was found to enhance the anti-bacterial effect of erythromycin, rifampicin, and vancomycin against the tested *Escherichia coli* strain [250].

#### 3.4.3. CBG Reduces the Expression of Biofilm and Quorum Sensing-Related Genes in *Streptococcus mutans*

CBG inhibited sucrose-induced biofilm formation by *Streptococcus mutans* with a minimum biofilm inhibitory concentration (MBIC) of 2.5 μg/mL [243]. Higher concentrations (10 μg/mL) of CBG were required to reduce the metabolic activity of preformed *Streptococcus mutans* biofilms [243]. CBG reduced the expression of various biofilm-related genes (e.g., *gtfB*, *gtfC*, *gtfD*, *ftf*, *gbpA*, *gbpA*, *brpA*, *wapA*) with concomitant reduction in the production of extracellular polymeric substances (EPS) [243]. The quorum sensing-related genes *comE*, *comD*, and *luxS* were downregulated by CBG, while no effect was observed on the gene expression of the stress-associated chaperones *groEL* and *dnaK* [243]. Moreover, CBG induced reactive oxygen species (ROS) production in *Streptococcus mutans*, which might be related to the reduced expression of the oxidative stress defense genes, *sod* and *nox* [243]. Thus, CBG has specific anti-biofilm activity unrelated to its membrane-acting effect. This conclusion is further supported by the study of Aqawi et al. [248] showing that CBG inhibited quorum sensing, bacterial motility, and biofilm formation of the marine Gram-negative *Vibrio harveyi* without affecting the planktonic growth.

#### 3.4.4. CBG and HU-210 Inhibit Quorum Sensing in *Vibrio harveyi*

Quorum sensing is an inter-bacterial communication system mediated by secreted autoinducers that interact with their respective receptors, resulting in the activation of a signal transduction cascade that alters the gene expression repertoire in a cell-density-dependent manner [274]. CBG prevented the bioluminescence induced by the master quorum sensing regulator LuxR of *Vibrio harveyi* at a concentration of 1 µg/mL [248]. Using a Δ*luxM*, Δ*lusS Vibrio harveyi* mutant that does not produce autoinducers AI-1 and AI-2, CBG was found to prevent the signals delivered by exogenously added autoinducers, with a more profound inhibitory effect on the AI-2-induced than on the AI-1-induced bioluminescence [248]. Further studies show that CBG prevented the expression of several quorum sensing genes in *Vibrio harveyi*, including *luxU*, *luxO*, *qrr1–5*, and *luxR*, which can explain the inhibitory effect of CBG on LuxR-mediated bioluminescence [248]. Altogether, these data demonstrate that CBG can interfere with bacterial quorum sensing.

The synthetic cannabinoid HU-210, which is a dimethylheptyl analog of Δ^8^-THC (Figure 1) and acts as a high-affinity CB1 and CB2 agonist [275,276], has been shown to inhibit quorum sensing in the *Vibrio harveyi* AI-1^−^, AI-2^+^ BB152 mutant, but it had barely any effect on the wild-type bacteria or the AI-1^+^, AI-2^−^ MM30 mutant [249]. This suggests that HU-210 specifically antagonizes the AI-2 pathway [249]. The concentration of HU-210 required to achieve the anti-quorum sensing activity was relatively high (20–200 µg/mL) [249], which is 2–3 magnitudes higher than that of CBG [248]. HU-210 prevented biofilm formation of the AI-1^−^, AI-2^+^ BB152 mutant with a BIC_50_ of 2 µg/mL and MBIC_90_ of 200 µg/mL, while no significant effect was seen on biofilm formation by the wild-type bacteria or the AI-1^+^, AI-2^−^ MM30 mutant [249]. However, the motility of *Vibrio harveyi* was reduced in all three strains at both 20 and 200 µg/mL HU-210 [249]. Gene expression studies showed that HU-210 at a concentration of 2 µg/mL reduced the expression of the master regulator *luxR* in both wild-type and AI-1^−^, AI-2^+^ BB152 strain, while it had no effect on the AI-1^+^, AI-2^−^ MM30 *Vibrio harveyi* mutant strain [249]. The *luxM* gene that encodes for AI-1 was upregulated by HU-210 [249].

## 4. Endocannabinoids

The endocannabinoid system (ECS) modulates many physiological processes, including the cardiovascular, gastrointestinal and immune systems, pain, learning, memory, perception, mood, appetite, metabolism, emotions, and sleep [22,112,113,277,278,279,280,281,282,283,284,285]. The bioactive endocannabinoid lipid mediators have potent anti-inflammatory activities [286,287,288,289,290,291]. In addition, they promote neural progenitor cell proliferation and differentiation, and have neuroprotective effects [20,292,293,294]. The effect on neural cell proliferation is mediated by both the CB1 and CB2 receptors [293,295,296].

### 4.1. The Endocannabinoid System

The endocannabinoid system is composed of: (1) the lipid active endogenous ligands *N*-arachidonoylethanolamine (anandamide; AEA) and 2-arachidonoylglycerol (2-AG); (2) their biosynthetic enzymes (e.g., diacylglycerol lipases (DAGL), *N*-acyl-phosphatidylethanolamine phospholipase D-like esterase (NAPE-PLD), and Ca^2+^-dependent and Ca^2+^-independent *N*-acetyltransferases); (3) their degradative enzymes (e.g., fatty acyl amide hydrolase (FAAH) and monoacylglycerol lipase (MAGL)); and (4) the CB1 and CB2 cannabinoid receptors [15,297,298]. The precursors of endocannabinoids (e.g., *N*-acyl-phosphatidylethanolamine (NAPE) and phosphatidylinositol-4,5-bisphosphate (PIP2)) are present in the lipid membranes, and the endocannabinoids are produced upon demand, usually after activation of certain G-protein-coupled receptors (GPCRs) and in response to an increase in the intracellular calcium levels [299,300,301,302].

### 4.2. The Production of AEA and 2-AG

The production of endocannabinoids requires one or two enzymatic steps, followed by their release into the extracellular space. AEA is usually produced from *N*-arachidonoyl-phosphatidylethanolamine phospholipid, and 2-AG is produced primarily from membrane phospholipid 1-stearoyl-2-arachidonoyl-sn-glycerol [297]. The synthesis of 2-AG involves two steps: first, the hydrolysis of its precursor phospholipid by a phospholipase (PLCβ, PLCγ2, or PLCε), followed by further cleavage by diacylglycerol lipase (DAGL) [303,304,305]. The biosynthesis of these endocannabinoids occurs in areas of the brain functionally related to cognitive processes, motivation, and movement control [306,307]. 2-AG was found to be present at 170 times higher concentrations than AEA in brain lysate [308]. While AEA was initially detected in the brain [135] and 2-AG in the canine gut [309], today it is known that these host-derived endocannabinoid lipid hormones are found in various peripheral tissues (e.g., the intestine) and in the serum, and produced by certain immune cells [23,290,309,310,311,312,313,314,315,316,317]. For instance, lipopolysaccharides induced the production of AEA in adipose tissue macrophages [318]. T and B cells produce elevated levels of 2-AG upon activation [290]. Astrocytes were found to produce AEA, as well as homo-γ-linolenoylethanolamine (HEA), docosatetraenoylethanolamine (DEA), oleoylethanolamine (OAE), and palmitoylethanolamine (PEA) [319].

### 4.3. The Circulating Levels of AEA and 2-AG

The circulating endocannabinoid levels are affected by various factors, and under physiological conditions, the AEA serum level was found to be between 1 to 5 nM, and the 2-AG serum level between 10–500 nM [316,320]. Physical exercise mobilizes endocannabinoids, which could contribute to the analgesic and mood-elevating effects of exercise [316]. The circulating levels of 2-AG show a circadian rhythm that gets altered when sleep is disrupted [316,320]. CBD inhibits the degradation of AEA and 2-AG, which is associated with the anti-inflammatory and anti-oxidative activities [321].

### 4.4. Endogenous Receptors for AEA and 2-AG

AEA and 2-AG act as agonists of the CB1 and CB2 receptors [135,322,323,324,325]. While 2-AG binds with high affinity to CB1 and CB2 cannabinoid receptors, AEA binds with low affinity to these receptors [323,324]. Although phytocannabinoids and endocannabinoids bind to the same CB1 and CB2 receptors, their chemical structure is quite different [297] (Figure 1 and Figure 3). Both AEA and 2-AG have an alkyl-amide (alkamide) chemical structure, while cannabinoids are terpenophenolic compounds.

In addition to acting on CB1 and CB2, AEA activates the ionotropic TRPV1 channel, resulting in the opening of the ion channel and Ca^2+^ influx [312,326,327,328,329,330,331], the G-protein-coupled receptor GPR55 [332,333], and the cation channel TRPA1 [334], while it inhibits the TRPM8 channel [334]. In addition, AEA activates PPARγ, and 2-AG activates PPARα [335]. The vasodilation action of AEA was found to be mediated via activation of TRPV1 [336]. Endocannabinoids activating TRPV1 have been included in the endovanilloid system [337,338,339]. Recent studies suggest that potassium channels are also the targets of endocannabinoids [340].

In the brain, endocannabinoids serve as retrograde synaptic messengers [299,341]. They are released from postsynaptic neurons and inhibit the release of presynaptic neurotransmitters, such as glutamate and gamma-aminobutyric acid (GABA) by binding to the CB1 receptor and TRPV1 expressed in the presynaptic terminals [299,342,343]. This has led to the hypothesis that endocannabinoids regulate over-excitability and promote synaptic homeostasis [344]. Endocannabinoids differ from the classical neurotransmitters in that they are not stored in vesicles but are released immediately after their production.

The solubility of endocannabinoids is low in water, raising the question of how AEA diffuses through the synaptic cleft [345]. There is evidence that AEA can interact with cholesterol and ceramide, which are required for their insertion into and transport through the membrane [345,346,347]. In the brain, the lipid-binding protein α-synuclein is involved in the transport of arachidonic acid [348]. Fatty acid binding proteins have been shown to be intracellular carriers of AEA [349].

Another communication system that exists between neurons is the release of lipid-based transport systems such as exosomes from neurons following a synaptic response, that are taken up by neighboring cells [350,351]. Gabrielli et al. [352] observed that endocannabinoids are secreted on extracellular membrane vesicles. In this study, extracellular vesicles secreted by microglial cells were found to carry AEA on their surface that was able to stimulate the CB1 receptor expressed on neurons and inhibit presynaptic transmission [352]. Microglial cells release endocannabinoids at much higher levels than neurons and astrocytes [319,353,354] and are thought to play a role in regulating the synaptic activity by a process termed gliotransmission, which functions to bridge the non-synaptic inter-neuronal communication [355].

### 4.5. Other Endocannabinoids and Endocannabinoid-like Compounds

Other endocannabinoids include the oleoyl- and palmitoyl-ethanolamines (OEA and PEA) that affect intestinal permeability by acting on TRPV-1 and PPARα [356,357], and 2-AG-ether and O-arachidonoylethanolamine (virodhamine) [22,358] (Figure 3). PEA is produced by neurons, microglia, and astrocytes in the central nervous system [359,360] where it plays an important role in neuroprotection [361,362]. Moreover, it was shown to have both anti-nociceptive and anti-inflammatory activities [363,364,365,366]. Immune cells release PEA that activates the CB2 receptor, resulting in downregulation of the inflammatory processes [367,368]. PEA, which is synthesized along with AEA, potentiates the action of AEA by increasing receptor affinity or reducing the degradation of AEA by FAAH [357,369,370,371]. The study of Lo Verme et al. [372] showed that PPARα was required for the anti-inflammatory effect of PEA. Borrelli et al. [365] observed that PEA alleviates the inflammation in a murine colitis model through acting on CB2, GPR55, and PPARα. OEA acts on PPARα and is secreted in the proximal intestine where it controls appetite, exhibits anti-inflammatory properties, and stimulates lipolysis and fatty acid oxidation [373,374,375,376].

The endocannabinoid noladin ether acts on CB2 and inhibits the intracellular effector adenylyl cyclase [377]. The endocannabinoid virodhamine, which is composed of arachidonic acid and ethanolamine joined by an ester linkage, is a partial agonist with an antagonist activity on CB1, while being a full agonist on CB2 [378]. At low concentrations, virodhamine activates GPR55, while at high concentrations it acts as an antagonist [379]. The endocannabinoid *N*-arachidonoyl-dopamine (NADA), which is highly expressed in the striatum, hippocampus, and cerebellum, activates TRPV1, induces the release of substance P and calcitonin gene-related peptide from dorsal spinal cord slices, and enhances hippocampal paired-pulse depression [380]. NADA and its epoxide metabolites also act as an agonist for the CB1 and CB2 receptors and show anti-inflammatory activities [337,381,382,383]. Other dopamine-related endocannabinoids include *N*-oleoyldopamine (OLDA), *N*-palmitoyldopamine (PALDA), and *N*-stearoyldopamine (STEARDA) [384]. OLDA is only a weak ligand of CB1, but it induced calcium influx, reduced the latency of paw withdrawal from a radiant heat source, and produced nocifensive behavior [384].

*N*-Arachidonoyl-L-serine (AraS) is an endogenous bioactive lipid found both in the central nervous system (CNS) and in the periphery, with a similar structure and physiological functions as AEA [385,386] (Figure 3). It possesses vasoactive, pro-angiogenic, pro-neurogenic, and neuroprotective properties [386,387,388]. Since AraS binds weakly to CB1 and CB2, it is not classified as an endocannabinoid, but rather has been coined as an “endocannabinoid-like” substance [386]. The pro-angiogenic activity of AraS is achieved by activation of GPR55 [387]. Moreover, AraS stimulates phosphorylation of MAPK and Akt protein kinases [385].

### 4.6. Anti-Microbial Activities of Endocannabinoids and Endocannabinoid-like Compounds 

The anti-microbial effect of endocannabinoids depends on the strain studied and the endocannabinoid used [16,17,18,389,390] (Table 2). Among the tested organisms, *Streptococcus salivarius*, *Bacteroides fragilis*, and *Enterococcus faecalis* were the most susceptible bacteria to AEA and *N*-Linoleoylethanolamine (LEA) [390]. MSSA and MDRSA become immediately growth arrested by AEA, an effect that was transient and relieved upon time [16]. On the other hand, the growth of *Lactobacillus gasseri* species becomes enhanced by LEA and OEA [390].

#### 4.6.1. AEA and AraS Exert Bacteriostatic Activity on Both Drug-Sensitive and Drug-Resistant *Staphylococcus aureus*

Feldman et al. [18] observed that the MIC of AEA toward three MRSA species (MRSA ATCC 33592, MRSA ATCC 43300, and a MRSA clinical isolate) was above 256 µg/mL. AraS had a MIC of 16 and 128 µg/mL on MRSA ATCC 33592 and MRSA ATCC 43300, respectively, and a MIC above 256 µg/mL for the clinical MRSA isolate [18]. A kinetic study of AEA on a multidrug-resistant *Staphylococcus aureus* (MDRSA) clinical isolate and the MSSA ATCC 25923 strain showed that AEA caused a transient bacteriostatic effect that was overcome with time [16]. The bacteriostatic effect of AEA was independent of the drug-resistant phenotype [16]. Further analysis showed that AEA inhibited cell division just prior to daughter cell separation [16]. Gene expression studies showed that AEA reduced the expression of some autolysin genes, which might in part contribute to the growth arrest [16]. AEA altered the membrane structure of the MDRSA and caused an immediate membrane depolarization that recovered with time [16]. Both AEA and AraS reduced the hydrophobicity index of MRSA at a concentration of 16 µg/mL [18].

#### 4.6.2. AEA and AraS Sensitize Drug-Resistant *Staphylococcus aureus* to Antibiotics

Importantly, it was observed that AEA and AraS sensitize MRSA and MDRSA strains to various antibiotics, including β-lactam antibiotics (ampicillin and methicillin), gentamicin, tetracycline, and norfloxacin [16,17]. For instance, the MIC of ampicillin against MRSA ATCC 33592 and ATCC 43300 was 128 and 256 µg/mL, respectively, but in the presence of 8–16 µg/mL AEA, it was reduced to 8 µg/mL [17]. The MIC of gentamicin against MRSA ATCC 33592 was 128 µg/mL, but in the presence of 8 µg/mL AEA, it was reduced to 4 µg/mL [17]. Treating a MDRSA clinical isolate with 50 µg/mL AEA reduced the MIC of methicillin from above 500 µg/mL to 50 µg/mL [16]. AEA was found to prevent drug efflux, resulting in intracellular drug accumulation, which might explain, at least in part, the sensitization of the bacteria to antibiotics [16]. Gene expression analysis shows that AEA reduces the expression of some efflux pump genes, including *norB*, *norC*, *mepA*, *kdpA*, and *opp1C* in MDRSA [16], but it is likely that the alterations in the membrane structure caused by AEA also contribute to intracellular drug retention.

It is notable that the sensitization of MRSA to methicillin takes place even when bacterial growth is inhibited by AEA [16], suggesting that the anti-bacterial effect of methicillin and other β-lactams does not require cell division as previously documented when used as a single agent [391,392]. Indeed, FtsZ inhibitors that arrest bacterial cell growth, also sensitize drug-resistant *Staphylococcus aureus* to β-lactam antibiotics, which was related to membrane relocalization of penicillin-binding proteins (PBPs) [393]. Further studies are required to fully understand the antibiotic-sensitization mechanisms of AEA and AraS.

#### 4.6.3. AEA and AraS Exhibit Anti-Biofilm Activity against Drug-Sensitive and Drug-Resistant *Staphylococcus aureus*

AEA and AraS prevent biofilm formation of MRSA and MDRSA with a maximum effect at 12.5–35 µg/mL [16,18]. AEA and AraS had a rather weak effect on preformed biofilm of MRSA and MDRSA, where concentrations as high as 64 µg/mL were required to eradicate 50% of the biofilms after a 24 h incubation [16,18]. The simultaneous treatment of the MRSA and MDRSA strains with endocannabinoids and antibiotics significantly lowered the effective dose of the two compounds [16,17]. For instance, the MBICs of AEA and ampicillin on MRSA ATCC 33592 were, respectively, 33.8 and 128 µg/mL, while in combination, 8 µg/mL of each compound was required for inhibiting biofilm formation [17]. When combining the sub-MBIC concentration 3.125 µg/mL of AEA with 50 µg/mL norfloxacin, which, as a single agent, had no anti-biofilm effect, an 80% reduction in biofilm formation by MDRSA was observed [16]. A 90% reduction in preformed MDRSA biofilm was observed when 50 µg/mL of AEA was combined with 50 µg/mL methicillin, which is the synergistic condition required for killing the bacteria [16]. The latter observation shows that the combination of AEA with antibiotics is also effective against biofilm-embedded bacteria.

Gene expression studies showed that AEA reduced the expression of the regulatory *RNAIII* and the virulence gene α-helical phenol-soluble modulin (*psmα*) in MDRSA [16]. Additionally, the genes *fnbB*, *hla*, and *hld* encoding for the virulence factors fibronectin binding protein, α-hemolysin and δ-hemolysin (δ-toxin), respectively, were downregulated by AEA [16]. As Psmα plays a central role in *Staphylococcus aureus* biofilm formation by stabilizing the biofilms through amyloid formation [394,395,396], the inhibition of its expression might be one mechanism for the anti-biofilm effect of AEA (Figure 4).

#### 4.6.4. AEA and AraS Inhibit Yeast-Hypha Transition of *Candida albicans* and Prevent Adhesion of *Candida albicans* Hyphae to Epithelial Cells

AEA and AraS were found to inhibit yeast-hypha transition of *Candida albicans* at 125 and 250 µg/mL [389]. At 50 µg/mL, there was only a partial inhibition on the yeast-hypha transition, but this concentration was sufficient to prevent hyphal extension [389]. Importantly, *Candida albicans* hyphae that have been exposed to AEA at 50 µg/mL and higher concentrations showed strong reduction in their ability to adhere to the HeLa cervical epithelial carcinoma cells [389]. AraS-treated *Candida albicans* hyphae showed deficient adherence to HeLa cervical carcinoma cells similar to AEA-treated fungi, while 2-AG treatment had only a minor effect at the concentrations analyzed (up to 250 µg/mL) [389]. None of the endocannabinoids affected the adherence of the *Candida albicans* hyphae to polystyrene tissue culture plates within the first hour of incubation, while AraS and 2-AG, but not AEA, reduced the biofilm mass formed on the polystyrene tissue culture plates after a 24 h incubation [389].

Gene expression studies showed that AEA increased the expression of *NRG1*, which is a transcriptional repressor of filamentous growth, but reduced the expression of the hyphal cell wall protein 1 (*HWP1*), the Agglutinin-like protein 3 (*ALS3*), the Hypha-specific G1 cyclin-related protein 1 (*HGC1*), the Ras-like protein 1 (*RAS1*), the enhanced filamentous growth protein 1 (*EFG1*), the cell surface hydrophobicity-associated protein *CSH1*, and the extent of cell elongation protein 1 (*ECE1*). The combined effect of AEA on the expression of these genes might cumulate in the observed effects of AEA on *Candida albicans* adherence and hyphal growth (Figure 5).

### 4.7. Dialog between the Gut Microbiota and the Endocannabinoid System

An intercommunication system has been found to exist between the gut microbiota and the endocannabinoid system [285,298,397,398,399,400,401,402]. The gut microbiota, representing more than 100 trillion microorganisms, including at least 1000 distinct species, lives in symbiosis with the host and assists in controlling the metabolic health of the host by degrading nutrients that the host is unable to digest and by providing a whole battery of small signaling molecules, metabolites, and nutrients beneficial for the host physiology [298,403]. The gut microbiome differs from individual to individual, and the composition of the microbiota is believed to affect various metabolic disorders, such as obesity, hyperglycemia, and dyslipidemia, which are risk factors for type 2 diabetes, hepatosteatosis, and arteriosclerosis [404]. Reduced diversity of gut microbiota has been linked with various pathophysiological conditions, such as depression, schizophrenia, neurological disorders, and chronic fatigue [405,406,407,408,409]. Microbiota can affect the endocannabinoid system and the nervous system, and vice versa; the nervous system and the endocannabinoids can influence the enteric microbiota composition [112,285,390,397,410,411].

#### 4.7.1. The Relationship between Gut Microbiota, the Endocannabinoid System, and Depression

The effect of the gut microbiota on depressive-like behaviors in mice was found to be mediated by the endocannabinoid system [411]. These authors showed that the transfer of microbiota from stress-induced depressive mice to naïve unstressed hosts induced a depressive-like state in the recipients. This was accompanied by a reduced adult hippocampal neurogenesis that was related to decreased hippocampal 2-AG levels and deficient CB1-mediated activation of the mTOR signaling pathways [411]. The detrimental effects on hippocampal neurogenesis could be restored by a MAGL inhibitor that prevents 2-AG degradation, addition of the 2-AG precursor arachidonic acid to the diet, or by complementation with *Lactobacillus plantarum*^WJL^ [411]. The microbiota from the stress-induced depressive mice showed an increase in *Ruminococcaceae* and *Porphyromonodaceae* species, with a decrease in *Lactobacillaceae* [411]. The complementation with *Lactobacillus plantarum*^WJL^ restored hippocampal 2-AG to normal levels, as well as increased the levels of AEA, n-3, and n-6 polyunsaturated fatty acids (PUFAs) [411]. *Lactobacillus* species can regulate fatty acid metabolism, absorption, and fatty acid composition of the host [412,413], which in turn affects the endocannabinoid system [411]. Rousseaux et al. [414] observed that oral administration of *Lactobacillus acidophilus* induced the expression of both the µ_1_ opioid receptor and CB2 receptor on colon epithelial cells, resulting in reduced abdominal pain in a rat colorectal distension model.

#### 4.7.2. Association between Gut Microbiota, PEA, and Anhedonia/Amotivation

PEA was shown to mediate the association between gut microbial diversity and anhedonia/amotivation [410]. Increased serum levels of PEA were associated with anti-depressive effects [357,415], while increased stool levels of PEA, indicative of increased excretion of PEA, were associated with alterations in synaptic plasticity, learning, and emotional responses [410,416,417]. The stool PEA levels were associated with gut microbial diversity, with implications on host mental health [285,357,410,418]. The relative abundance of microbes of the *Blautia* and *Dorea* taxa was particularly associated with fecal PEA and anhedonia/amotivation [410]. Reduced microbial diversity corresponded with increased excretion of PEA and more severe anhedonia/amotivation [410]. PEA was also found to counteract autistic-like behaviors in BTBR T^+^ *tf*/J mice by dampening inflammation, reducing oxidative stress, reducing gut permeability, and altering the gut microbiota, besides its neuroprotection through induction of PPARα [419]. These authors found that PEA treatment increased the ratio of *Firmicutes*/*Bacteroidetes*, which was due to an increase in *Firmicutes* (e.g., *Clostridials*) and a decrease in *Bacteroidetes* [419].

#### 4.7.3. The Relationship between Gut Microbiota, AEA, and Acute Respiratory Distress Syndrome

AEA was found to attenuate acute respiratory distress syndrome through modulating the gut microbiota [420]. In this study, the researchers investigated the effect of AEA on staphylococcal enterotoxin B (SEB)-mediated acute respiratory distress syndrome. SEB caused an increase in pathogenic bacteria in both the lungs and the gut [420]. AEA-treated mice showed increased level of anti-microbial peptides in the lung epithelial cells and prevented the increase in pathogenic bacteria induced by SEB [420]. AEA increased the level of several bacterial species (e.g., *Lachnospiraceae* and *Clostridia*) that produce elevated levels of SCFAs, such as butyrate and valerate, important for stabilizing the gut–lung microbial axis and suppressing inflammation [420]. In addition, AEA treatment increased the abundance of *Muribaculaceae* and reduced the abundance of *Pseudomonas* and *Enterobacteriaceae* [420].

#### 4.7.4. The Relationship between Gut Microbiota, the Endocannabinoid System, and Obesity

Obesity is often characterized by low-grade inflammation, with increased levels of endocannabinoids in the plasma and adipose tissues and altered expression of CB1 [421]. Activation of CB1 and CB2 receptors reduces motility, limits secretion, and decreases hypersensitivity in the gut [422,423,424]. Impaired CB1 signaling protected against the development of obesity and steatosis [425,426,427]. Lipopolysaccharides from Gram-negative bacteria induce the production of endocannabinoids under inflammatory conditions that dampen the inflammatory response [318,428,429,430]. Vice versa, the activation of CB1 in mice increases circulating levels of lipopolysaccharides due to reduced expression of the tight junction proteins occludin and zonula occludens-1 (ZO-1), resulting in increased gut epithelial permeability [421,431].

The gut microbiota was found to modulate colon CB1 receptor expression in both normal and obese mice [421]. Obese mice fed with the prebiotic oligofructose showed reduced CB1 expression, lower AEA content, and increased expression of FAAH [421]. Obese mice treated with the CB1 antagonist SR141716A (Rimonabant) improved the gut barrier function and reduced body weight gain [421]. Mehrpouya-Bahrami et al. [432] observed that SR141716A attenuated diet-induced obesity and inflammation that was correlated with increased relative abundance of *Akkermansia muciniphila* and decreased abundance of *Lanchnospiraceae* and *Erysipelotrichaceae* in the gut. Interestingly, SR141716A prevented the intracellular replication of macrophage-phagocytosed *Brucella suis* by activating the macrophages, which was related to the inhibition of CB1 [433]. It would be interesting to study whether SR141716A also has a direct anti-microbial effect. In this context, it is worth mentioning that SR141716A could potentiate the anti-fungal activity of amphotericin B against *Candida albicans* and *Cryptococcus neoformans* by increasing cellular oxidative stress and cell membrane permeability [434].

Mice fed on high-fat, high-glucose diet showed altered microbiome with concomitant increase in AEA and 2-AG in the plasma [398]. The relative abundances of *Adlercreutzia*, *Barnesiella*, *Coprobacillus*, *Eubacterium*, and *Parasutterella* in the ileum were negatively associated with AEA levels [398]. The level of the AEA congener *N*-docosahexaenoylethanolamine (synaptamide, DHEA), which is required for normal brain development [435], was negatively associated with *Barnesiella*, *Enterococcus*, *Eubacterium*, *Flavonifractor*, and *Intestinimonas* in the ileum [398]. These authors also found a negative correlation between the *Delftia* genus and *N*-linoleoylethanolamine (LEA), while the *Lactobacillus* genus was associated with increased 2-docosahexaenoyl-glycerol (2-DHG) levels [398].

Repeated administration of OAE to mice fed on normal chow pellet diet for 11 days led to alteration in fecal microbial composition with an increase in *Bacteroidetes* (e.g., *Bacteroides* genus) and a decrease in *Firmicutes* (*Lactobacillus*), which is considered a “lean-like” phenotype [436]. OAE also reduced intestinal cytokines expression by immune cells isolated from Peyer’s patches [436].

#### 4.7.5. The Relationship between Gut Microbiota, the Endocannabinoid System, and Inflammatory Bowel Diseases

Both AEA and PEA have been observed to reduce inflammation in murine models of colitis and inflammatory bowel disease [437,438,439]. Elevating the levels of the endocannabinoids by inhibiting FAAH could relieve colitis and inflammatory bowel disease [440,441]. Butyrate that is produced by gut microbiota (e.g., bacteria of the *Ruminococcaceae* and *Lachnospiraceae* family) [442] reduces inflammation and pain in colitis animal models, which can in part be mediated through the endocannabinoid system [443]. Vijay et al. [443] studied the association of the endocannabinoids AEA, 2-AG, OEA, and PEA with gut microbiome composition upon exercise. Under resting condition, AEA and OEA were positively associated with alpha diversity and with SCFA producing bacteria such as *Bifidobacterium*, *Coprococcus 3*, and *Faecalibacterium*, while being negatively associated with *Collinsella* [443]. AEA, OEA, and PEA increased with exercise, and changes in AEA correlated with bacterial butyrate production [443]. The increases in AEA and PEA correlated with decreased expression of the inflammatory mediators TNFα and IL-6 and increased expression of the anti-inflammatory cytokine IL-10 [443].

PEA was found to increase the phagocytosis and intracellular killing of encapsulated *Escherichia coli* K1 by activated microglial cells and macrophages [444,445]. Pre-treatment with PEA significantly increased the survival of mice challenged with *Escherichia coli* K1 [445]. Similarly, Heide et al. [446] observed that prophylactic PEA attenuated inflammation and increased the survival of mice challenged with intracerebral *Escherichia coli* K1 infection. Lower bacterial loads were observed in the spleen, liver, and blood of the PEA pretreated animals [446]. This was related to the anti-inflammatory effect, since PEA at 1 μg/mL had no effect on *Escherichia coli* growth in vitro [446].

The gut microbiota of IBD patients differs from healthy individuals, with a decrease in butyrate- and indole-producing bacteria, decrease in bile salt-sensitive bacteria, while an enrichment in bile acid-metabolizing bacteria [390,447,448,449]. Among others, *Escherichia coli*, *Lactobacillus gasseri*, *Ruminococcus gnavus*, and *Blautia producta*, were more abundant in IBD, while *Bacteroides cellulosilyticus*, *Bacteroides fragilis*, and *Streptococcus salivarius* were depleted [390]. Fornelos et al. [390] observed that certain *N*-acylethanolamines (NAEs), such as LEA, PEA, OEA and AEA, are elevated in the stool of IBD, Crohn’s disease, and/or ulcerative colitis patients, and stimulate the growth of bacterial species overrepresented in IBD while inhibiting bacterial species lacking in IBD. *N*-acylethanolamine levels were highest in samples with most differences in the microbiome, suggesting a connection between *N*-acylethanolamines and altered microbiota in IBD [390]. These researchers observed that LEA inhibited the growth of *Bacteroides fragilis*, *Bacteroides cellulosilyticus*, and *Enterococcus faecalis*, while slightly enhanced the growth of *Escherichia coli*, *Ruminococcus gnavus*, and *Blautia producta* [390]. AEA also transiently inhibited the growth of *Bacteroides fragilis* and *Enterococcus faecalis* that recovered with time [390]. The growth of *Lactobacillus gasseri*, which is enriched in IBD, was enhanced by OEA and LEA, and to a lesser extent by AEA [390]. LEA and AEA partly inhibited the growth of *Alistipes shahii* and *Ruminococcus lactaris* that are underrepresented in IBD subjects [390]. The growth of *Streptococcus salivarius* was completely prevented at a concentration of 50 μM LEA, AEA, or OEA, but only slightly inhibited by PEA [390].

#### 4.7.6. Effect of *N*-acylethanolamines on the Microbial Composition of Stool Chemostats

Fornelos et al. [390] also studied the effect of *N*-acylethanolamines, including endocannabinoids, on the composition of two different stool chemostats. In the control chemostat A, the *Enterobacteriaceae*, *Clostridiaceae*, and *Veillonellaceae* taxa dominated, while in the presence of AEA, the bacterial community was almost entirely overtaken by *Enterobacteriaceae* [390]. LEA-treated chemostat A was dominated by *Enterococcaceae*, *Veillonellaceae* and *Enterobacteriaceae* at the expense of *Streptococcaceae*, *Erysipelotrichaceae*, *Porphyromonadacea*, *Bacteroidaceae*, and *Rikenellaceae*, while OEA treatment did not impact *Enterobacteriaceae* abundance but increased the relative abundances of *Enterococcaceae* and *Streptococcaceae* and decreased those of *Bacteroidaceae* and *Rikenellaceae* [390]. AEA and LEA also reduced the abundance of *Barnesiella intestinihominis*, *Alistipes*, and *Bacteroides* species, while they increased the abundance of *Escherichia* species [390]. In Chemostat B, the abundance of *Blautia producta*, *Clostridium clostridioforme*, *Klebsiella pneumoniae*, and *Proteus mirabilis* was increased in the presence of AEA or LEA [390]. These data indicate that LEA, AEA, and other *N*-acylethanolamines can shift the microbiome of a healthy individual into an IBD profile [390].

Transcriptional analysis showed that AEA upregulates both the anaerobic, reductive, and oxidative branches of the citrate cycle concomitant with increased energy metabolism and increased respiratory electron transport chain activity, especially in *Enterobacteriaceae* [390]. Metabolic changes occurring upon exposure to AEA, LEA, and other NAEs include the activation of bacterial processes involved in NAE metabolism [390].

Searching for an action mechanism of LEA and AEA on *Bacteroides fragilis*, the researchers found that the most upregulated genes are those encoding for membrane-associated efflux transport proteins, and the most downregulated gene was the long chain fatty acid (LCFA) importer *fadL* [390]. They further showed that two other genes involved in fatty acid metabolic processing were repressed: a *fadD* homolog that catalyzes esterification of incoming fatty acids into CoA thioesters and a *fadE* homolog involved in downstream fatty acid breakdown [390]. These data indicate that bacteria have developed mechanisms that can respond to endocannabinoids.

## 5. Conclusions

Both phytocannabinoids and endocannabinoids have diverse physiological activities that are, in part, mediated by common receptors in mammalians, each compound with its specificity and affinity, being agonists, partly agonists, inverse agonists, or antagonists. While these mechanisms have been widely investigated, the mechanisms leading to their anti-microbial effects are less understood. Despite the quite different structures of phytocannabinoids and endocannabinoids (Figure 1 and Figure 3), there are some common dominators that characterize their anti-bacterial activities (Figure 2 and Figure 4). For instance, they exert bacteriostatic activity, alter the membrane structure, induce either membrane hyperpolarization or depolarization, modulate gene expression including those involved in metabolism, affect virulence factors, and prevent biofilm formation (Figure 2 and Figure 4). Additionally, the AEA-mediated inhibition of yeast-hypha transition of *Candida albicans* and the hyphal adherence to epithelial cells seem to be mediated by alterations in gene expression (Figure 5). The multiple actions of phytocannabinoids and endocannabinoids suggest that the compounds do much more than just affecting membrane permeability as previously thought.

The emergence of antibiotic-resistant microbes is a clinical problem worldwide, and novel treatment strategies are urged. The important observation that some of the phytocannabinoids and endocannabinoids act on both drug-sensitive and drug-resistant *Staphylococcus aureus* makes them potential antibiotic adjuvants in treating drug-resistant infections, for instance for topical infectious wound treatment. Especially important is the ability of AEA and AraS to sensitize drug-resistant *Staphylococcus aureus* to antibiotics in virtue of their ability to prevent drug efflux and induce growth arrest. The addition of these endocannabinoids to the treatment cocktail will revive the use of already existing antibiotics. The ability of both CBG and the synthetic cannabinoid HU-210 to antagonize quorum sensing may have implications in the new era where quorum sensing inhibitors or quorum quenchers have attracted attention for alternative antibiotic drugs for antibiotic-resistant bacteria [450]. To reach this goal, further studies should be performed to clarify the spectrum of bacteria whose quorum sensing pathways are affected by cannabinoids. The increasing recognition of the complex interplay between the gut microbiota and the endocannabinoid system with the accompanying implications for various physiological and pathophysiological conditions, places the therapeutic uses of cannabinoids into a new spotlight.

## Figures and Tables

**Figure 1 biomedicines-10-00631-f001:**
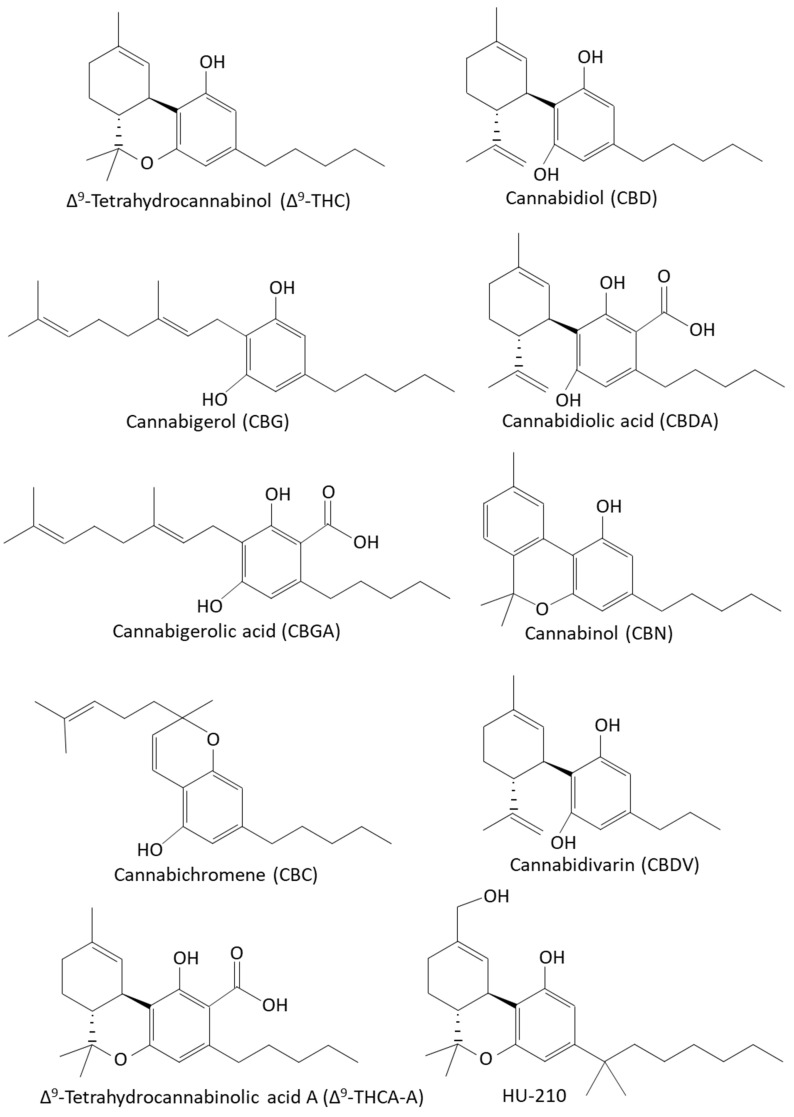
The chemical structures of some phytocannabinoids and the synthetic cannabinoid HU-210.

**Figure 2 biomedicines-10-00631-f002:**
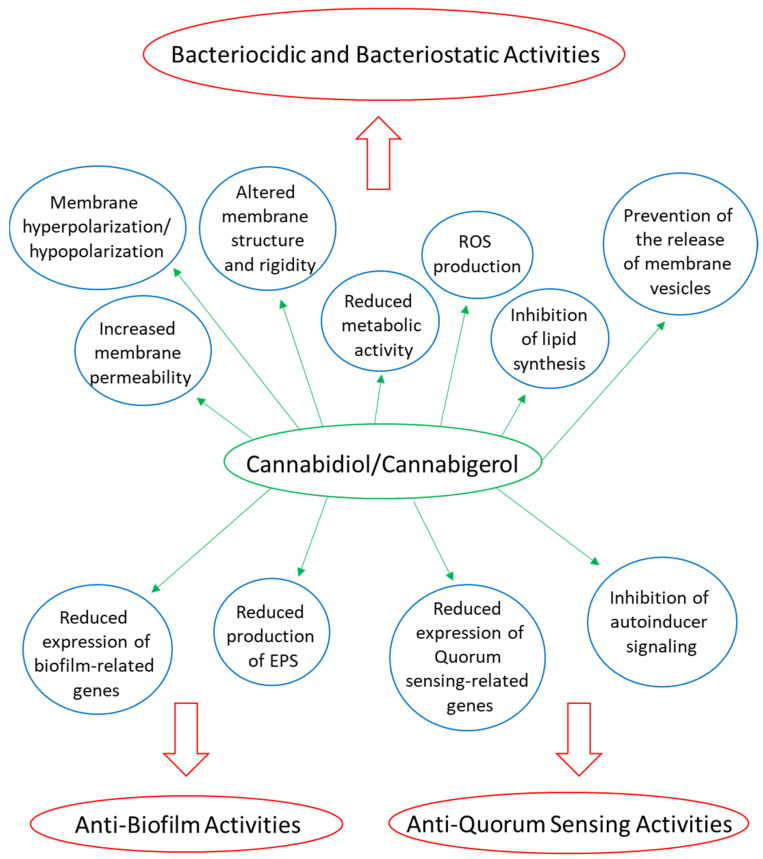
The anti-bacterial activities of phytocannabinoids.

**Figure 3 biomedicines-10-00631-f003:**
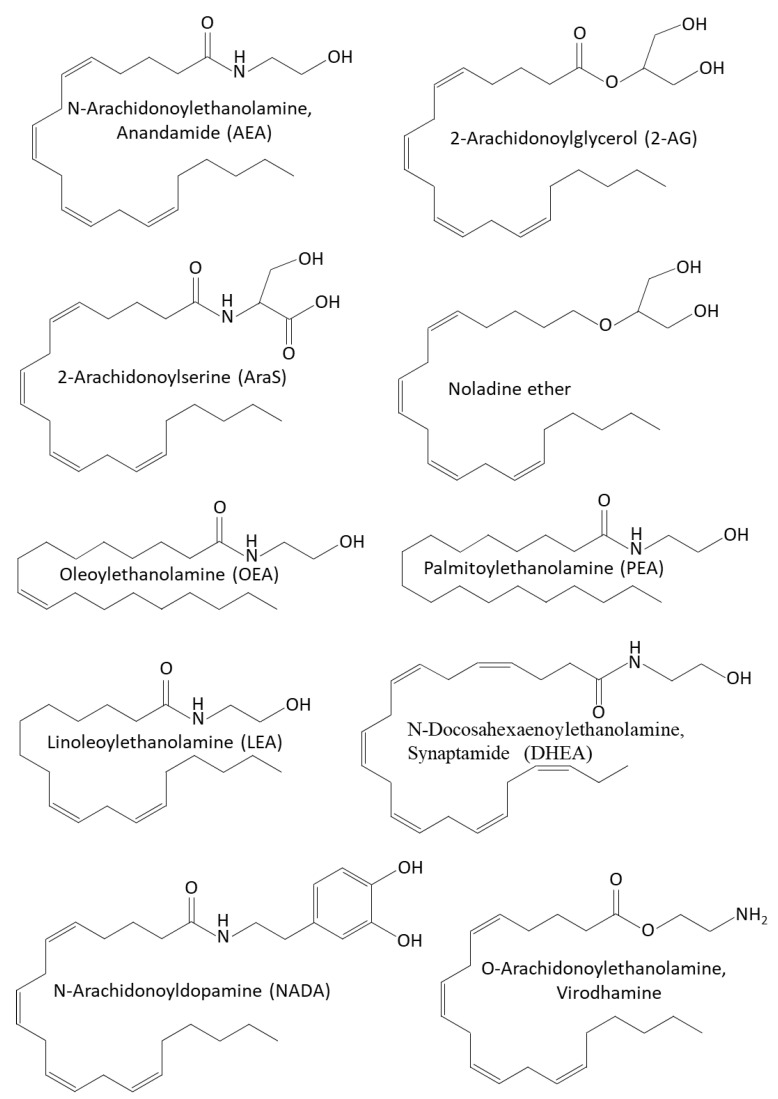
The chemical structures of some endocannabinoids.

**Figure 4 biomedicines-10-00631-f004:**
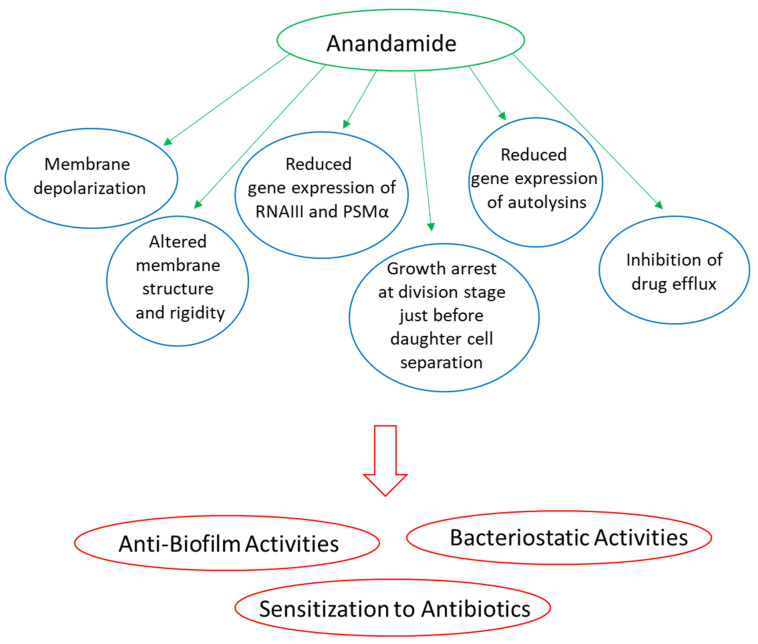
The anti-bacterial actions of anandamide on *Staphylococcus aureus*.

**Figure 5 biomedicines-10-00631-f005:**
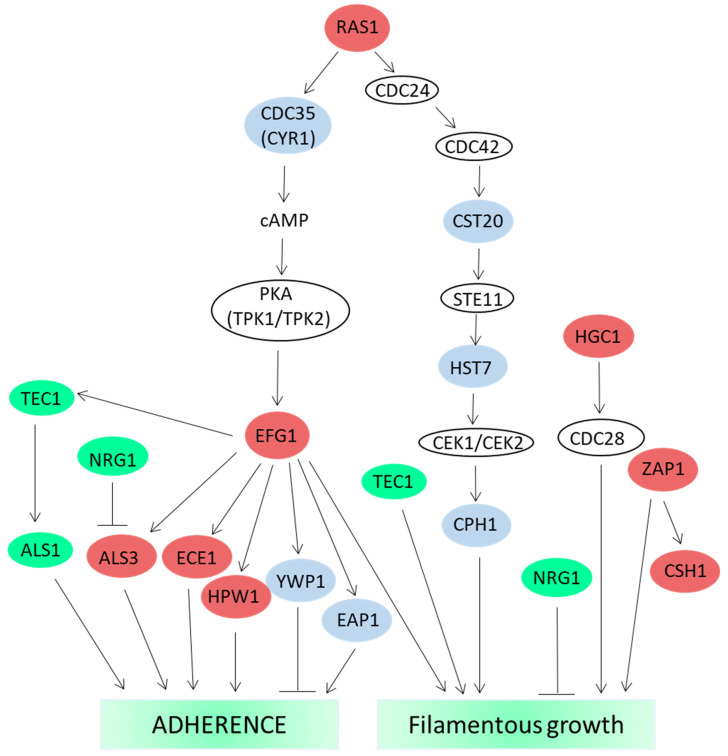
Effect of AEA on *Candida albicans* gene expression [381]. Genes in red are downregulated by AEA after a 2 h incubation. Genes in green are upregulated by AEA. Genes in light blue are unaffected by AEA. Open circles are genes that have not yet been analyzed. Ultimately, AEA prevents the adherence and hyphal extension of *Candida albicans*.

**Table 1 biomedicines-10-00631-t001:** Examples of *Cannabis sativa* constituents that have been documented to possess anti-bacterial, anti-fungal, and/or anti-protozoal activities *.

Phytocannabinoids	Anti-Microbial Activity	Reference
**∆^9^-Tetrahydrocannabinol (Δ^9^-THC)**	MIC: 2–5 μg/mL against *Staphylococcus aureus* ATCC 6538 MIC: 1 μg/mL against *Staphylococcus aureus* ATCC 25923 MIC: 2 μg/mL against *Staphylococcus aureus* SA-1199B (NorA overexpression) MIC: 2 μg/mL against *Staphylococcus aureus* EMRSA-15 MIC: 0.5 μg/mL against *Staphylococcus aureus* EMRSA-16 MIC: 2 μg/mL against MRSA USA300 MIC: 4–8 μg/mL against MRSA ATCC 43300 MIC: 5 μg/mL against *Streptococcus pyogenes* MIC: 2 μg/mL against *Streptococcus milleri* MIC: 5 μg/mL against *Streptococcus faecalis* MIC: 4–8 μg/mL against *Neisseria gonorrhoeae* ATCC 19424 IC_50_: 4.8 μM against *Staphylococcus aureus* ATCC 29213 IC_50_: 6.9 μM against *Bacillus cereus* IIIM 25 IC_50_: 2.8 μM against *Lactococcus lactis* MTCC 440 IC_50_: 3.5 μM against *Shigella boydii* NC-09357 IC_50_: 6.4 μM against *Staphylococcus warneri* MTCC 4436 No effect against *Escherichia coli*, *Salmonella typhi* or *Proteus vulgaris*	[58,61,245,246,247]
**Cannabidiol (CBD)**	MIC: 1–5 μg/mL against *S. aureus* ATCC 6538 MIC: 0.5–1 μg/mL against *Staphylococcus aureus* ATCC 25923 MIC: 1 μg/mL against *Staphylococcus aureus* SA-1199B (NorA overexpression) MIC: 1 μg/mL against *Staphylococcus aureus* EMRSA-15 MIC: 1 μg/mL against *Staphylococcus aureus* EMRSA-16 MIC: 1–4 μg/mL against MRSA USA300 MIC: 1–2 μg/mL against various *Staphylococcus aureus* isolates. MIC: 1–2 μg/mL against *Staphylococcus epidermidis*. MIC: 4 μg/mL against methicillin-resistant *Staphylococcus epidermidis*. MIC: 2 μg/mL against *Streptococcus pyogenes* MIC: 1 μg/mL against *Streptococcus milleri* MIC: 5 μg/mL against *Streptococcus faecalis* MIC: 1–4 μg/mL against various *Streptococcus pneumoniae* species MIC: 0.5–4 μg/mL against various *Enterococcus faecalis* species MIC: 4 μg/mL against *Listereria monocytogenes* MIC: 1–2 μg/mL against *Cutibacterium (Propionibacterium) acnes* ATCC 6919 MIC: 2–4 μg/mL against *Clostridioides (Clostridium) difficile* M7404 human ribotype 027 MIC: 1–2 μg/mL against various *Neisseria gonorrhoeae* isolates. MIC: 0.25 μg/mL against various *Neisseria meningitidis* ATCC 13090 MIC: 1 μg/mL against *Moraxella catarrhalis* MMX 3782 MIC: 1 μg/mL against *Legionella pneumophila* MMX 7515 IC_50_: 3.8 μM against *Staphylococcus aureus* ATCC 29213 IC_50_: 9.5–11.1 μM against *Staphylococcus aureus* ATCC 6538 IC_50_: 9.8 μM against *Bacillus cereus* IIIM 25 IC_50_: 2.9 μM against *Lactococcus lactis* MTCC 440 IC_50_: 4.3 μM against *Shigella boydii* NC-09357 IC_50_: 4.1 μM against *Pseudomonas fluorescens* MTCC 103 IC_50_: 5.7 μM against *Staphylococcus warneri* MTCC 4436 Moderate effect against *Mycobacterium smegmatis* (MIC 16 μg/mL) and marginal activity against *Mycobacterium tuberculosis* H37Rv, *Candida albicans*, and *Cryptococcus neoformans* with a MIC > 64 μg/mL. No effect against *Escherichia coli*, *Salmonella typhimurium, Shigella dysenteriae, Proteus vulgaris*, *Proteus mirabilis*, *Klebsiella pneumoniae*, *Pseudomonas aeruginosa*, *Acinetobacter baumannii*, *Serratia marcescens*, *Burkholderia cepacian*, and *Haemophilus influenzae*. *Anti-biofilm effect*: MBEC: 1–4 μg/mL against MSSA and MRSA biofilms. BIC_50_: 12.5 μg/mL against *Candida albicans* SC5314 MBIC: 100 μg/mL against *Candida albicans* SC5314	[58,61,62,245,246,247,251,252,253]
**Cannabigerol (CBG)**	MIC: 0.5 μg/mL against *Staphylococcus aureus* ATCC 25923 MIC: 1 μg/mL against *Staphylococcus aureus* SA-1199B (NorA overexpression) MIC: 2 μg/mL against *Staphylococcus aureus* EMRSA-15 MIC: 1 μg/mL against *Staphylococcus aureus* EMRSA-16 MIC: 2 μg/mL against MRSA USA300 MIC: 2–4 μg/mL against various MRSA clinical isolates, with some requiring > 8 μg/mL MIC: 4–8 μg/mL against MRSA ATCC 43300 MIC: 2.5 μg/mL against *Streptococcus mutans* UA159 ATCC 700610 MIC: 1 μg/mL against *Streptococcus sanguis* ATCC 10556 MIC: 5 μg/mL against *Streptococcus sobrinus* ATCC 27351 MIC: 5 μg/mL against *Streptococcus salivarius* ATCC 25975 MIC: 1–2 μg/mL against *Neisseria gonorrhoeae* ATCC 19424 IC_50_: 15 μg/mL against *Mycobacterium intracellulare* *Anti-biofilm effect*: MBIC: 2–4 μg/mL against biofilm formation by MRSA 4 μg/mL eradicated preformed biofilms of MRSA MBIC: 2.5 μg/mL against biofilm formation by *Streptococcus mutans* UA159 ATCC 70061 *Anti-quorum sensing effect* 1 μg/mL CBG inhibited quorum sensing in *Vibrio harveyi* BB120.	[58,61,100,243,244,247,248]
**Cannabidiolic acid (CBDA)**	MIC: 1–2 μg/mL against *Neisseria gonorrhoeae* ATCC 19424 MIC: 2 μg/mL against *Staphylococcus aureus* ATCC 25923 MIC: 4 μg/mL against *Staphylococcus aureus* USA300 MIC: 4 μg/mL against *Staphylococcus epidermidis* CA#71 and ATCC 51625 MIC: 16–32 μg/mL against MRSA ATCC 43300 No effect on *Escherichia coli* ATCC 25922 or *Pseudomonas aeruginosa* PA01 with a MIC > 64 μg/mL.	[62,247]
**Cannabigerolic acid (CBGA)**	IC_50_: 12 μg/mL against *Leishmania donovani* MIC: 4 μg/mL against MRSA USA300 MIC: 2–4 μg/mL against MRSA ATCC 43300 MIC: 1–2 μg/mL against *Neisseria gonorrhoeae* ATCC 19424	[61,100,247]
**Cannabichromene (CBC)**	MIC: 1.56 μg/mL against *Staphylococcus aureus* ATCC 6538 MIC: 2 μg/mL against *Staphylococcus aureus* ATCC 25923 MIC: 2 μg/mL against *Staphylococcus aureus* SA-1199B (NorA overexpression) MIC: 2 μg/mL against *Staphylococcus aureus* EMRSA-15 MIC: 2 μg/mL against *Staphylococcus aureus* EMRSA-16 MIC: 8 μg/mL against MRSA USA300 MIC: 0.39 μg/mL against *Bacillus subtilis* ATCC 6633 MIC 12.5 μg/mL against *Mycobacterium smegmatis* ATCC 607 IC_50_: 5.9 μM against *Staphylococcus aureus* ATCC 29213 IC_50_: 9.2 μM against *Bacillus cereus* IIIM 25 IC_50_: 2.6 μM against *Lactococcus lactis* MTCC 440 IC_50_: 3.4 μM against *Shigella boydii* NC-09357 IC_50_: 5.6 μM against *Staphylococcus warneri* MTCC 4436	[58,61,220,246]
**Cannabichromenic acid (CBCA)**	MIC: 2 μg/mL against MRSA USA300 MIC: 7.8 μM against *Staphylococcus aureus* MSSA 34397 MIC: 3.9 μM against a clinical MRSA isolate MIC: 7.8 μM against vancomycin-resistance *Enterococcus faecalis* (VRE)	[61,254]
**Cannabinol (CBN)**	MIC: 1 μg/mL against *Staphylococcus aureus* ATCC 25923 MIC: 1 μg/mL against *Staphylococcus aureus* SA-1199B (NorA overexpression) MIC: 1 μg/mL against *Staphylococcus aureus* EMRSA-15 MIC: 2 μg/mL against MRSA USA300 IC_50_: 7.9 μM against *Staphylococcus aureus* ATCC 29213 IC_50_: 3.2 μM against *Bacillus cereus* IIIM 25 IC_50_: 5.8 μM against *Lactococcus lactis* MTCC 440 IC_50_: 11.7 μM against *Shigella boydii* NC-09357 IC_50_: 8.3 μM against *Pseudomonas fluorescens* MTCC 103 IC_50_: 9.2 μM against *Staphylococcus warneri* MTCC 4436	[58,61,246]
**Cannabidivarin (CBDV)**	MIC: 2–4 μg/mL against MRSA ATCC 43300 MIC: 0.03–0.5 μg/mL against *Neisseria gonorrhoeae* ATCC 19424 IC_50_: 7.8 μM against *Staphylococcus aureus* ATCC 29213 IC_50_: 3.1 μM against *Bacillus cereus* IIIM 25 IC_50_: 3.2 μM against *Lactococcus lactis* MTCC 440 IC_50_: 10.4 μM against *Shigella boydii* NC-09357 IC_50_: 5.9 μM against *Pseudomonas fluorescens* MTCC 103 IC_50_: 7.9 μM against *Staphylococcus warneri* MTCC 4436 IC_50_: 11.9 μM against *Candida albicans* MTCC 4748 MIC: 8 μg/mL against MRSA USA300	[61,246,247]
**(-)Δ^8^-Tetrahydrocannabinol** **(Δ^8^-THC)**	MIC: 2 μg/mL against MRSA USA300 MIC: 4–8 μg/mL against MRSA ATCC 43300 MIC: 2–4 μg/mL against *Neisseria gonorrhoeae* ATCC 19424	[61,247]
**Exo-tetrahydrocannabinol (exo-THC)**	MIC: 2 μg/mL against MRSA USA300	[61]
**Δ^9^-Tetrahydrocannabinolic** **acid A (THCA-A)**	MIC: 4 μg/mL against MRSA USA300	[61]
**Δ^9^-Tetrahydrocannabivarin (THCV)**	MIC: 4 μg/mL against MRSA USA300 MIC: 64 μg/mL against MRSA ATCC 43300 MIC: 16 μg/mL against *Neisseria gonorrhoeae* ATCC 19424	[61,247]
**Δ^1^-Tetrahydrocannabidivarol**	IC_50_: 6.9 μM against *Staphylococcus aureus* ATCC 29213 IC_50_: 6.9 μM against *Bacillus cereus* IIIM 25 IC_50_: 5.1 μM against *Lactococcus lactis* MTCC 440 IC_50_: 3.9 μM against *Shigella boydii* NC-09357 IC_50_: 7.8 μM against *Pseudomonas fluorescens* MTCC 103 IC_50_: 7.6 μM against *Staphylococcus warneri* MTCC 4436	[246]
**(±)-4-Acetoxycannabichromene**	IC_50_: 40.3 μM against *Leishmania donovani* IC_50_: 4–7.2 μM against *Plasmodium falciparum*	[63]
**(±)-3″-Hydroxy-Δ^(4″,5″)^ cannabichromene**	IC_50_: 24.4 μM against MRSA ATCC 33591 IC_50_: 29.6 μM against *Staphylococcus aureus* ATCC 29213 IC_50_: 60.5 μM against *Candida albicans* ATCC 90028 IC_50_: 60.5 μM against *Candida krusei* ATCC 6258 IC_50_: 57.5 μM against *Leishmania donovani* Not active against *Escherichia coli, Mycobacterium intracellulare*, or *Plasmodium falciparum.*	[63]
**5-Acetyl-4-hydroxycannabigerol**	IC_50_: 53.4 μM against MRSA ATCC 33591 IC_50_: 10.7 μM against *Leishmania donovani* IC_50_: 6.7–7.2 μM against *Plasmodium falciparum* Not active against *Staphylococcus aureus, Escherichia coli, Mycobacterium intracellulare*, or *Candida albicans.*	[63]
**4-Acetoxy-2-geranyl-5-hydroxy-3-n-pentylphenol**	IC_50_: 6.7 μM against MRSA ATCC 33591 IC_50_: 12.2 μM against *Staphylococcus aureus* ATCC 29213 IC_50_: 53.4 μM against *Candida krusei* ATCC 6258 IC_50_: 42.7 μM against *Leishmania donovani* Not active against *Escherichia coli, Mycobacterium intracellulare, Candida albicans*, or *Plasmodium falciparum*.	[63]
**8-Hydroxycannabinol**	IC_50_: 4.6 μM against *Candida albicans* ATCC 90028 IC_50_: 30.6 μM against *Mycobacterium intracellulare* Not active against *Escherichia coli.*	[63]
**8-Hydroxycannabinolic acid A**	IC_50_: 54 μM against *Candida krusei* ATCC 6258 IC_50_: 3.5 μM against *Staphylococcus aureus* ATCC 29213 IC_50_: 54 μM against *Escherichia coli* Not active against *Mycobacterium intracellulare.*	[63]
**Non-Cannabinoid constituents of *Cannabis sativa* L.**		
**5-Acetoxy-6-geranyl-3-n-pentyl-1,4-benzoquinone**	IC_50_: 15 μg/mL against MRSA ATCC 43300 IC_50_: 13 μg/mL against *Leishmania donovani* IC_50_: 2.6–2.8 μg/mL against *Plasmodium falciparum*	[101]
**Cannflavin A**	IC_50_: 4.5 μg/mL against *Leishmania donovani*	[101]
**Cannflavin B**	IC_50_: 5 μg/mL against *Leishmania donovani*	[100]
**Cannflavin C**	IC_50_: 17 μg/mL against *Leishmania donovani*	[101]
**6-Prenylapigenin**	IC_50_: 6.5 μg/mL against MRSA ATCC 43300 IC_50_: 20 μg/mL against *Candida albicans* IC_50_: 2.0–2.8 μg/mL against *Plasmodium falciparum*	[101]
**Prenylspirodinone**	IC_50_: 49.6 μM against *Bacillus thuringiensis* MTCC 809	[246]

* BIC_50_ = The test concentration that prevents 50% biofilm formation compared to control cells. IC_50_ = The test concentration that causes 50% growth inhibition in comparison to control cells. MBEC = Minimum biofilm eradication concentration is the lowest concentration that completely eradicates preformed biofilm. MBIC = Minimum biofilm inhibitory concentration is the lowest concentration that is required to completely prevent any biofilm formation. MIC = Minimum inhibitory concentration is the lowest concentration that completely inhibits bacterial growth (when no turbidity is observed).

**Table 2 biomedicines-10-00631-t002:** Anti-microbial activities of endocannabinoids and endocannabinoid-like compounds.

Endocannabinoids	Anti-Microbial Activity	References
**Anandamide (AEA)**	MIC: 50 μM against *Streptococcus salivarius* RJX1086. MIC > 256 µg/mL against MSSA ATCC 25923, MRSA ATCC 33592, MRSA ATCC 43300, a MRSA clinical isolate, and a MDRSA clinical isolate. Transient bacteriostatic activity against drug-sensitive and drug-resistant *Staphylococcus aureus* species in a dose-dependent manner at concentration equal to an above 12.5 μg/mL AEA. 50 and 100 µM AEA slightly inhibited the growth of *Alistipes shahii* RJX1084 and *Ruminococcus lactaris* RJX1085, and caused a delay in the log-phase growth of *Bacteriocides fragilis* ATCC 25285. 100 µM AEA retarded the growth of *Enterococcus faecalis* RJX1251. 50 and 100 µM AEA slightly enhanced the growth of *Lactobacillus gasseri* DSM 20243, *Escherichia coli* RJX1083, and *Ruminococcus gnavus* RJX1118, while causing a small delay in the log-phase of *Ruminococcus gnavus* ATCC 29149. 50 µM, but not 100 µM, AEA slightly increased the growth of *Lactobacillus gasseri* RJX1262. 50 and 100 µM AEA had no effect on the growth of *Escherichia coli* AIEC NC101. AEA sensitizes MRSA and MRDSA to antibiotics, including β-lactam antibiotics (ampicillin and methicillin), gentamicin, tetracycline, and norfloxacin. MBIC: 12.5–35 µg/mL against MSSA ATCC 25923, MRSA ATCC 33592, MRSA ATCC 43300, a MRSA clinical isolate, and a MDRSA clinical isolate. No anti-biofilm effect against *Candida albicans*. Concentrations above 50 μg/mL prevented yeast-hyphal transition and hyphal extension of *Candida albicans* and inhibited their adhesion to cervical epithelial cells.	[16,17,18,389,390]
***N*-Arachidonoyl-L-serine (AraS)**	MIC: 16 µg/mL against MRSA ATCC 33592. MIC: 128 µg/mL against MRSA ATCC 43300. MIC > 256 µg/mL against a MRSA clinical isolate. AraS sensitizes MRSA to antibiotics, including β-lactam antibiotics (ampicillin and methicillin), gentamicin, and tetracycline. MBIC: 12.5–35 µg/mL against MRSA ATCC 33592, MRSA ATCC 43300, and a MRSA clinical isolate. MBIC_50_: 50 µg/mL against *Candida albicans*. Concentrations above 50 μg/mL prevented yeast-hyphal transition and hyphal extension of *Candida albicans* and inhibited their adhesion to cervical epithelial cells.	[17,18,389]
**2-Arachidonoylglycerol (2-AG)**	MBIC_50_: 125 µg/mL against *Candida albicans*.	[389]
***N*-Linoleoylethanolamine (LEA)**	MIC: 50 μM against *Streptococcus salivarius* RJX1086. MIC: 96 µM against *Bacteroides fragilis* ATCC 25285. A delayed log-phase growth was observed with 24 and 48 µM LEA on *Bacteroides fragilis* ATCC 25285. MIC: 100 µM against *Enterococcus faecalis* RJX1251, with a strong growth retardation with 50 µM. MIC: 100 µM against *Alistipes shahii* RJX1084. 50 and 100 µM LEA reduced the growth of *Ruminococcus lactaris* RJX1085 by 24–40%. 50 µM LEA strongly stimulated the growth of *Lactobacillus gasseri* DSM 20243 and *Lactobacillus gasseri* RJX1262. 100 µM LEA strongly stimulated the growth of *Lactobacillus gasseri* DSM 20243 but slightly interfered with the growth of *Lactobacillus gasseri* RJX1262. 50 and 100 µM LEA slightly increased the growth of *Escherichia coli* RJX1083. It had no effect on the growth of *Ruminococcus gnavus* ATCC 29149 at 100 µM, with a small delay in the log-phase growth at 200 µM. It had no effect on the growth of *Escherichia coli* AIEC NC101, even at 200 µM.	[390]
**Oleoylethanolamine (OEA)**	MIC: 50 μM against *Streptococcus salivarius* RJX1086. 50 and 100 µM OEA had a slight growth inhibitory effect on *Ruminococcus lactaris* RJX1085 and caused a delay in the log-phase growth of *Alistipes shahii* RJX1084, *Bacteroides fragilis* ATCC 25285, and *Enterococcus faecalis* RJX1251. 50 and 100 µM OEA strongly increased the growth of *Lactobacillus gasseri* RJX1262, while 100 µM was required to stimulate the growth of *Lactobacillus gasseri* DSM 20243 and *Ruminococcus gnavus* RJX1118. Both 50 and 100 µM had a slight growth-stimulating effect on *Escherichia coli* RJX1083, while no significant effect was observed on *Escherichia coli* AIEC NC101 and *Ruminococcus gnavus* ATCC 29149.	[390]
**Palmitoylethanolamine (PEA)**	50 and 100 µM PEA partly reduced the growth of *Ruminococcus lactaris* RJX1085 and *Streptococcus salivarius* RJX1086. The effect on *Alistipes shahii* RJX1084 was subtle. 100 µM, but not 50 µM, PEA enhanced the growth of *Bacteroides fragilis* ATCC 25285 and *Enterococcus faecalis* RJX1251. 50 and 100 µM PEA slightly increased the growth of *Lactobacillus gasseri* RJX1262 and DSM 20243, *Escherichia coli* RJX1083, *Ruminococcus gnavus* ATCC 29149 and RJX1118, while it had no significant effect on *Escherichia coli* AIEC NC101.	[390]

## Data Availability

Not applicable.

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
