# Peer review of "Anti-Microbial Activity of Phytocannabinoids and Endocannabinoids in the Light of Their Physiological and Pathophysiological Roles"

_biomedicines, 2022, doi:10.3390/biomedicines10030631_

Round 1

Reviewer 1 Report

In this review paper, the authors discussed the general properties, anti-microbial and anti-film activities of Phytocannabinoids and endocannabinoids. The whole idea is very interesting. The manuscript is well organized and well-written. I only have two minor comments.

1, From 3.2-3.8, THC, CBD, CBG, CBC, CBDA, CBGA and CBN are all common Phytocannabinoids. So, it would be better to put them in the same subtitle, like 3.2.1-3.2.7.  

2, For 2-AG synthesis, recent studies found that except PLCβ, PLCγ and PLCε also regulated the synthesis of endocannabinoid. See publications: 10.1073/pnas.2112971118, PMC5354337

Author Response

We want to thank the reviewer for taking time to review the manuscript and to valuable comments.

In this review paper, the authors discussed the general properties, anti-microbial and anti-film activities of Phytocannabinoids and endocannabinoids. The whole idea is very interesting. The manuscript is well organized and well-written. I only have two minor comments.

1, From 3.2-3.8, THC, CBD, CBG, CBC, CBDA, CBGA and CBN are all common Phytocannabinoids. So, it would be better to put them in the same subtitle, like 3.2.1-3.2.7.

We have accordingly changed the numbers. This is indicated in red in the text. 

  1. For 2-AG synthesis, recent studies found that except PLCβ, PLCγ and PLCε also regulated the synthesis of endocannabinoid. See publications: 10.1073/pnas.2112971118, PMC5354337.

We have now included this information in the manuscript, outlined in red in the text.

Reviewer 2 Report

The authors have extensively collected the studies reported in the literature on the antimicrobial activity of phytocannabinnoids, endocannabinoids and some of their analogues. The article is well organized and clearly described.

Line 528– Please, also report in the text the following study: S Zaami, A Sirignano, Ó García-Algar, E Marinelli. COVID-19 pandemic, substance use disorders and body image issues, a worrisome correlation. Eur Rev Med Pharmacol Sci. 2022 Jan;26(1):291-297. doi: 10.26355/eurrev_202201_27780.

Line 601 – Please, also report in the text the following study: Sara Malaca, Francesco Paolo Busardò, Giulio Nittari, Ascanio Sirignano, Giovanna Ricci ,Fourth generation of synthetic cannabinoid receptor agonists: A review on the latest insights. Curr Pharm Des. 2021 Nov 15. doi: 10.2174/1381612827666211115170521.

Author Response

We want to thank the reviewer for taking time to review the manuscript and to provide valuable comments.

The authors have extensively collected the studies reported in the literature on the antimicrobial activity of phytocannabinnoids, endocannabinoids and some of their analogues. The article is well organized and clearly described.

  • Line 528– Please, also report in the text the following study: S Zaami, A Sirignano, Ó García-Algar, E Marinelli. COVID-19 pandemic, substance use disorders and body image issues, a worrisome correlation. Eur Rev Med Pharmacol Sci. 2022 Jan;26(1):291-297. doi: 10.26355/eurrev_202201_27780.

We have accordingly added this reference to the manuscript. This is outlined in blue in the text.

  • Line 601 – Please, also report in the text the following study: Sara Malaca, Francesco Paolo Busardò, Giulio Nittari, Ascanio Sirignano, Giovanna Ricci ,Fourth generation of synthetic cannabinoid receptor agonists: A review on the latest insights. Curr Pharm Des. 2021 Nov 15. doi: 10.2174/1381612827666211115170521.

We have accordingly added this reference to the manuscript. This is outlined in blue in the text.

Reviewer 3 Report

The present review focuses on the anti-microbial activities of phytocannabinoids and endocannabinoids interwoven with selected aspects of their many physiological and pathophysiological activities.

The study is very interesting and well described. But it still has some flaws.

If we check the title: Anti-Microbial Activity of Phytocannabinoids and Endocannabinoids in The Light of Their Physiological and Pathophysio
Logical Roles, why is there different activity such as anticancer anti-insect etc.

Authors mentioned several chemovariants, chemotypes or cultivars of Cannabis. These are very interesting and I wrote some more information about them.

I think that is better described firstly the activity of extracts and their antimicrobial activity, than essential oils and its activity against microorganisms.

The same is for Phytocannabinoids and Endocannabinoids. The manuscript idea is very good but authors need to find some system of description. 

Working in this order of description is quite confusing and authors should consider a different breakdown.

Author Response

We want to thank the reviewer for taking time to review the manuscript and to provide valuable comments.

The present review focuses on the anti-microbial activities of phytocannabinoids and endocannabinoids interwoven with selected aspects of their many physiological and pathophysiological activities.

The study is very interesting and well described. But it still has some flaws.

  • If we check the title: Anti-Microbial Activity of Phytocannabinoids and Endocannabinoids in The Light of Their Physiological and Pathophysiological Roles, why is there different activity such as anticancer anti-insect etc.

This review has tried to include the diverse activities of phytocannabinoids and endocannabinoids, and thus also included studies that have demonstrated anti-cancer and anti-insect activities. Indeed the cannabinoids have many other biological functions as stated by the reviewer. Since these are not the main focus of the manuscript, they are not included in the title of the manuscript.

  • Authors mentioned several chemovariants, chemotypes or cultivars of Cannabis. These are very interesting and I wrote some more information about them.

We have added two additional references describing this issue: Vergara et al. 2020 and  Borroto Fernandez et al., 2020. 

  • I think that is better described firstly the activity of extracts and their antimicrobial activity, than essential oils and its activity against microorganisms. The same is for Phytocannabinoids and Endocannabinoids. The manuscript idea is very good but authors need to find some system of description. Working in this order of description is quite confusing and authors should consider a different breakdown.

We believe that we have arranged the manuscript more and less as the reviewer has suggested. Each section begins with a general description, followed by physiological and pharmacological effects and then the anti-microbial activities. This flow scheme provides the readers essential background of the involved compounds and systems which is necessary for comprehending the sections describing the anti-microbial activities.

The review consists of three major sections that have been divided into subsections, which should give a reasonable flow:

  1. The Cannabis sativa L. plant.

- General aspects of the Cannabis sativa L. plant

- Anti-microbial activity of Cannabis sativa L. extracts

- Anti-microbial activity of essential oils from Cannabis sativa L.

  1. The Phytocannabinoids

- General aspects of phytocannabinoids.

- Pharmacological effects of phytocannabinoids.

- Anti-microbial effects of phytocannabinoids.

- Some mechanistic insight into the anti-bacterial activity of phytocannabinoids

  1. The Endocannabinoids

- General aspects of endocannabinoids.

- Anti-microbial activities of endocannabinoids.

- Dialogue between the gut microbiota and the endocannabinoid system.

Round 2

Reviewer 3 Report

Manuscript is very well described and author accepted all comments.